# Deciphering the Tangible Spatio-Temporal Spread of a 25-Year Tuberculosis Outbreak Boosted by Social Determinants

Mariana G. López,[a] Ma Isolina Campos-Herrero,[b] Manuela Torres-Puente,[a] Fernando Cañas,[c] Jessica Comín,[d] Rodolfo Copado,[e] Penelope Wintringer,[f] Zamin Iqbal,[f] Eduardo Lagarejos,[b] Miguel Moreno-Molina,[a] Laura Pérez-Lago,[g] Berta Pino,[h] Laura Sante,[i] Darío García de Viedma,[g,j] Sofía Samper,[d,j] Iñaki Comas[a,k]

aTuberculosis Genomics Unit, Instituto de Biomedicina de Valencia (IBV), CSIC, Valencia, Spain

bServicio de Microbiología, Hospital Universitario de Gran Canaria Dr. Negrín, Las Palmas de Gran Canaria, Spain

cHospital Universitario Insular de Gran Canaria, Las Palmas de Gran Canaria, Spain

dInstituto Aragonés de Ciencias de la Salud, Fundación IIS Aragón, Zaragoza, Spain

eHospital José Molina Orosa, Las Palmas de Gran Canaria, Spain

fEuropean Molecular Biology Laboratory – European Bioinformatics Institute, Hinxton, UK

gServicio Microbiología Clínica y Enfermedades Infecciosas, Hospital General Universitario Gregorio Marañón, Instituto de Investigación Sanitaria Gregorio Marañón, Madrid, Spain

hHospital Nuestra Señora de la Candelaria, Santa Cruz de Tenerife, Spain

iHospital Universitario de Canarias, Santa Cruz de Tenerife, Spain

jCIBER Enfermedades Respiratorias, Instituto de Salud Carlos III, Madrid, Spain

kCIBER Epidemiología y Salud Pública, Instituto de Salud Carlos III, Madrid, Spain

Darío García de Viedma and Sofía Samper contributed equally to this article.

**ABSTRACT** Outbreak strains of *Mycobacterium tuberculosis* are promising candidates as targets in the search for intrinsic determinants of transmissibility, as they are responsible for many cases with sustained transmission; however, the use of low-resolution typing methods and restricted geographical investigations represent flaws in assessing the success of long-lived outbreak strains. We can now address the nature of outbreak strains by combining large genomic data sets and phylodynamic approaches. We retrospectively sequenced the whole genome of representative samples assigned to an outbreak circulating in the Canary Islands (the GC strain) since 1993, which accounts for ~20% of local tuberculosis cases. We selected a panel of specific single nucleotide polymorphism (SNP) markers for an *in-silico* search for additional outbreak-related sequences within publicly available tuberculosis genomic data. Using this information, we inferred the origin, spread, and epidemiological parameters of the GC strain. Our approach allowed us to accurately trace the historical and more recent dispersion of the GC strain. We provide evidence of a highly successful nature within the Canarian archipelago but limited expansion abroad. Estimation of epidemiological parameters from genomic data disagree with a distinctive biology of the GC strain. With the increasing availability of genomic data allowing for the accurate inference of strain spread and critical epidemiological parameters, we can now revisit the link between *Mycobacterium tuberculosis* genotypes and transmission, as is routinely carried out for SARS-CoV-2 variants of concern. We demonstrate that social determinants rather than intrinsically higher bacterial transmissibility better explain the success of the GC strain. Importantly, our approach can be used to trace and characterize strains of interest worldwide.

**IMPORTANCE** Infectious disease outbreaks represent a significant problem for public health. Tracing outbreak expansion and understanding the main factors behind emergence and persistence remain critical to effective disease control. Our study allows researchers and public health authorities to use Whole-Genome Sequencing-based methods to trace outbreaks, and shows how available epidemiological information helps to evaluate the

Address correspondence to Sofía Samper, ssamper.iacs@aragon.es, or Iñaki Comas, icomas@ibv.csic.es.

The authors declare a conflict of interest. Iñaki Comas received consultancy fees from Foundation for innovative new diagnostics. The author has no other competing interests to declare.

factors underpinning outbreak persistence. Taking advantage of all the freely available information placed in public repositories, researchers can accurately establish the expansion of an outbreak beyond original boundaries, and determine the potential risk of a strain to inform health authorities which, in turn, can define target strategies to mitigate expansion and persistence. Finally, we show the need to evaluate strain transmissibility in different geographic contexts to unequivocally associate spread to local or pathogenic factors, an important lesson taken from genomic surveillance of SARS-CoV-2.

**KEYWORDS** tuberculosis, outbreak, whole-genome sequencing, phylodynamics, genomic epidemiology

The identification of intrinsically highly transmissible strains of *Mycobacterium tuberculosis* remains a challenging task. Candidates include those strains thriving in a community for decades and significantly contributing to the long-term local tuberculosis burden. These long-lived outbreak strains have been identified in different parts of the world, and it has been speculated a link between higher transmissibility and success. Several studies have attempted to analyze the epidemiological characteristics of these strains and their genomic composition to reveal potential determinants of transmission; however, these studies are usually confined to their original geographic boundaries. But a lesson learned from SARS-CoV-2 variants of concern is that these studies of transmissibility should be replicated in different parts of the world. Previous attempts have failed to examine the success of outbreak strains on a global scale; thus, it remains unknown whether long-lived outbreak strains possess a similar or different trajectory in other countries, casting doubts about their transmissibility level.

## ADDED VALUE OF THIS STUDY

We analyzed a *Mycobacterium tuberculosis* strain causing a long-lived outbreak in the Canary Islands (since 1993) using whole-genome sequencing (WGS). As in previous studies of similar outbreak strains, we analyzed the diversity and phylodynamics of the outbreak in the area where it was initially described; however, thanks to the possibility of interrogating the entire European Nucleotide Archive, we had the unique chance to evaluate strain spread beyond original geographic boundaries. This approach allowed us to comprehensively trace the spatiotemporal spread of the GC strain from the emergence of its ancestor about 700 years ago to its recent transmission outside the Canary Islands. As a result, we found evidence for a limited success of the GC strain outside the Canary Islands. Furthermore, we combined our analysis with epidemiological data of early cases and phylodynamic analysis to estimate critical epidemiological parameters linked to strain spread. All evidence strongly suggests that factors related to the host, instead of the bacteria, support the persistence and expansion of this outbreak strain.

## IMPLICATIONS OF ALL THE AVAILABLE EVIDENCE

Infectious disease outbreaks are a major problem for public health. Monitoring outbreak expansion and understanding the main factors behind emergence and persistence remain essential to effective disease control. In this regard, it has been commonly proposed that outbreaks are caused by intrinsically more transmissible strains, against other factors like social or host determinants of disease; however, this has never been tested. We use WGS-based methods combined with available epidemiological information to evaluate the factors underpinning outbreak persistence. Taking advantage of all the freely available genomic information placed in public repositories, we demonstrate the power of our comprehensive approach to establish the expansion of an outbreak beyond original boundaries, and determine the potential risk of a strain. Finally, we provide significant evidence supporting the success of Gran Canaria outbreak-associated strain, and probably others, relates to ecological factors associated with founder effects linked to the host and social determinants (1) of disease.

According to the WHO, tuberculosis (TB) has surpassed HIV infection to become the

leading cause of death by an infectious disease before COVID-19. The year 2019 saw a reported 10 million new TB cases and 1.4 million deaths (2), with numbers likely to increase due to the COVID-19 pandemic (3). Outbreaks, defined as the concentration of an abnormal number of disease cases in space and time, are widespread in TB and ascribed to a single genotype or strain (1). While outbreaks have generally been assumed to be short-lived, genotyping has identified outbreak strains that become highly prevalent in specific regions and undergo transmission over decades. Outbreak strains can be found worldwide and across all lineages of the *Mycobacterium tuberculosis* complex (MTBC); furthermore, they are usually associated with local TB burden (1, 4–8). However, the success of outbreak strains has been primarily evaluated only in their original location, raising the question of whether these strains have any intrinsic transmissibility advantage or if their success derived from local population processes such as founder effects or ecological drivers of transmission (1, 4–8).

In this study, we investigated a TB outbreak strain (the Gran Canaria or GC strain), and compared this case to other previously published cases, to understand more about the origin and epidemiology of long-lived strains. A TB outbreak was identified in the Canary Islands in 2001; 651 strains were retrospectively analyzed between 1991 and 1996 in Gran Canaria, (the most populated island of the Canarian archipelago) using IS*6110*-restriction fragment length polymorphisms (RFLP). A significant cluster of 75 isolates was recognized, with the first 10 cases diagnosed in 1993. The likely index case was a Liberian refugee who arrived on the island 6 months before diagnosis in July of 1993 (9). The cluster was caused by a single strain, named GC1237, which belongs to the Beijing genotype (as corroborated by spoligotyping). More recently, 3 Gran Canaria outbreak-associated single nucleotide polymorphisms (SNPs) were selected and used to design a specific PCR to quickly identify secondary cases belonging to the Gran Canaria outbreak (10). Various molecular typing methods demonstrated the rapid spread of the GC1237 strain to the other islands of the Canarian archipelago, and identified new cases outside the archipelago up to 2014 (10–12). Until now, the Gran Canaria outbreak had not been studied using WGS.

In this work, we redefined the outbreak taking advantage of the increased resolution provided by WGS. In addition, we tracked the outbreak-associated strain within its original geographic boundaries and elsewhere by querying the entire European Nucleotide Archive (ENA) database. We combined epidemiological and sequencing data, and applied a phylodynamic analysis to trace the origin, track the spread, understand the dynamics, and define the factors that underpin the expansion of the outbreak strain.

## RESULTS

**Genomic redefinition off the outbreak.** We retrieved 86 samples from Gran Canaria outbreak patients previously typified by molecular methods, plus 3 additional samples from the likely index case collected over different years. We discarded 24 samples that did not achieve sufficient DNA quality for sequencing, leaving us with 65 usable samples (62 patients, ~10% of the outbreak) (Table S1). The outbreak was redefined following the diagram detailed in Fig. 1. First, we queried all sequences for the 3 SNPs previously defined as Gran Canaria outbreak markers (partial SNP profile) (10). We observed that 2 isolates (GC466 and GC515), considered part of the outbreak based on molecular typing methods, did not harbor those marker SNPs; thus, we excluded both from further analysis. In addition, we queried our collection of sequences for the partial SNP profile, and found 7 additional samples, 2 from the Valencia Region and 5 from Liberia (Table S2). We then constructed a maximum likelihood (ML) tree with all the samples harboring the partial SNP profile. We observed a differentiated and well-supported monophyletic clade (bootstrap = 99) (Fig. 2A), including the index case, most of the cases previously assigned to the outbreak, and the Valencian isolates (60 single TB cases) (Table S1). This clade likely represents the most precise genomic delimitation of the outbreak based on WGS data, and is referred to as "GC outbreak" hereafter, in order to indicate the outbreak caused by this particular genotype (GC strain). All samples included in the GC outbreak are genomically linked, and most cases are geographically and epidemiologically linked. The phylogeny also revealed that the SNPs

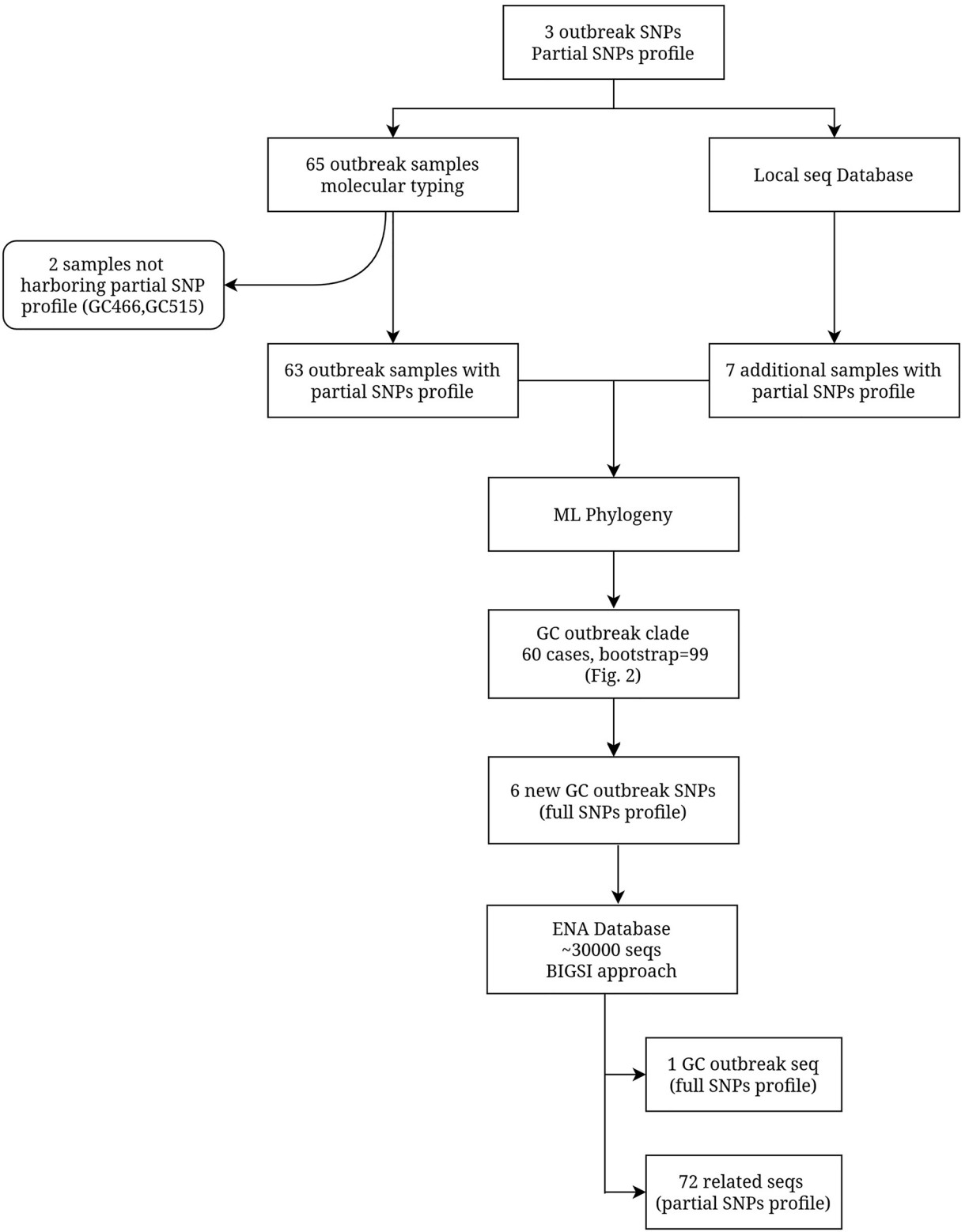

**FIG 1** Workflow detailing the GC outbreak delineation procedure.

initially considered as Gran Canaria outbreak-specific, were not exclusive, given that the additional global strains also harbor these SNPs.

We estimated the pairwise distance among samples included in the GC outbreak (Table 1), to evaluate if distances agreed with the accepted recent transmission thresholds of 0 to 10

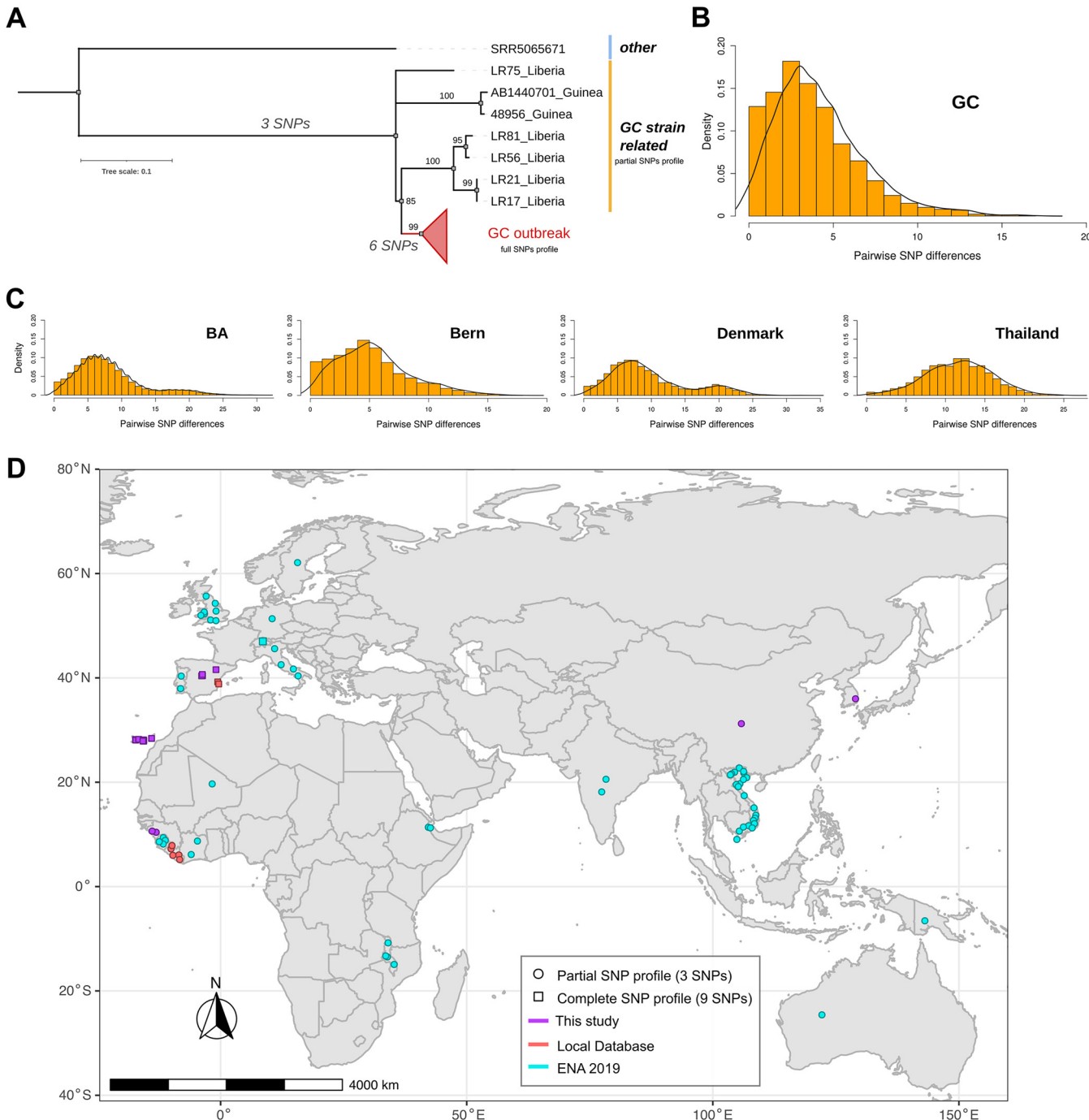

**FIG 2** GC outbreak characterization and sequence distribution (A) ML tree highlights the outbreak's circumscription and related strains identified with the partial SNP profile. (B and C) Density graphs of the pairwise number of SNPs between samples of different outbreaks Gran canaria (GC); Buenos Aires (BA) (5); Bern (6); Denmark (1); Thailand (4). (D) Distribution of sequences belonging to the outbreak (squares) and related to GC strain (circles). Colors indicate the source and shapes denote the meeting SNP profile of each sequence included in the study. The map was obtained from the R package *rnaturalearth* (https://docs.ropensci.org/rnaturalearth/articles/rnaturalearth.html).

SNPs. We found mean and median distance within-GC outbreak of 4.3 and 4 SNPs (range 0 to 17), respectively, and distance values between GC outbreak and non-outbreak samples of 33 and 30 SNPs (range 18 to 63).

Compared to other known outbreak-causing strains, the GC strain displayed the lowest within-outbreak mean and median pairwise distance, with values similar to the Bern outbreak (Table 1, and Fig. 2B and C). The GC and Bern outbreaks exhibited a unimodal right-skewed pairwise genetic distance distribution, indicating that most

**TABLE 1** Pairwise distance comparison among different outbreaks[a]

| Outbreak | SNP within-outbreak pairwise distance | | | | | |
|---|---|---|---|---|---|---|
| | Min | Max | Q1 | Median | Mean | Q2 |
| GC | 0 | 17 | 2 | 4 | 4.3 | 6 |
| BA | 0 | 31 | 5 | 7 | 8.2 | 10 |
| Bern | 0 | 18 | 3 | 5 | 5.4 | 7 |
| Denmark | 0 | 33 | 6 | 8 | 9.9 | 13 |
| Thailand | 0 | 25 | 9 | 12 | 11.8 | 15 |

[a]Minimum (Min), maximum (Max), mean, median, first (Q1) and second (Q2) quartile pairwise distance are provided.

of the samples have low pairwise SNP distance. Meanwhile, Denmark and Buenos Aires (BA) displayed a slightly bimodal distribution, suggesting the existence of more than one distinctive clade within both outbreaks, as proposed elsewhere (1). The Thailand outbreak displayed a unimodal normal distribution indicating a high pairwise SNP dispersion.

Thus, distance values obtained for the GC strain also support the genomic redefinition of the GC outbreak. With this delimitation, we excluded 2 additional samples from Madrid (AB1440701 and 48956), belonging to recent immigrants from Guinea, and previously identified as being infected by the GC1237 strain (10) (Table S1).

**Tangible GC strain spread and success outside the Canary Islands.** To understand the presence of the GC strain outside the Canary Islands and whether it displays similar epidemiological success when present, we designed an approach to evaluate all MTBC sequences available in public databases. This allowed us to study the GC strain's spread within the study area and inspect its broader extension. First, the genomic redefinition identified 6 additional markers of the GC outbreak, most of which were missense mutations located in genes involved in metabolic processes and respiration, with effects likely unrelated to virulence or transmission (Table S3 and Fig. 2A). Thus, we used the complete SNP profile (9 SNPs) to query every *M. tuberculosis* sample deposited in the ENA by July 2018 using the BIGSI index (13) (see Materials and Methods). We identified 1 sequence meeting the complete SNP profile from Switzerland, which was thus considered part of the GC outbreak, and 72 additional sequences harboring only the partial SNP profile, which are thus related to the GC strain (Table S2 and Fig. 2D). We considered these sequences in further analyses, as they may shed light on the remote origin of the GC strain.

**GC outbreak topology.** To study the epidemiological characteristics of the GC outbreak, we constructed a median-joining network with all samples, including the 3 additional samples from the likely index case (Table S1 and Fig. 3). The genomic network displays a star-like structure. The likely index case locates to the center (including the oldest sample and 2 additional samples collected in 1994 and 1995), along with 9 other isolates (most of them from 1993, and all from Gran Canaria). An in-depth investigation of the central node (Node A) revealed 1 heterogeneous SNP (hSNP) at genomic position 3190007, for which the first sample of the likely index case (GC077) harbors both the reference (T: 82.7%) allele, shared by the rest of samples of the outbreak, and the alternative allele (C: 17.3%), which is fixed in sample GC092 from 1994 (Fig. 3). The other hSNPs identified were uninformative since only 1 allele was fixed, either the reference or the alternative.

Connected with Node A, we observed 3 additional smaller star-like structures, corresponding to Nodes B, C, and E (Fig. 3) resembling secondary outbreaks. Node B represents the later spreading (around 2008) from Gran Canaria to the other islands in the Canarian archipelago, with an additional independent introduction in Tenerife. Node C, the smallest secondary outbreak, also occurred in Gran Canaria. Notably, the most prominent secondary outbreak, Node E, started early and included samples from the Spanish peninsula (Madrid and Zaragoza). Furthermore, the network revealed that the index case initiated a new transmission chain in 1997 (isolate GC1441), also in Gran Canaria. Surprisingly, the 2 samples from Valencia were directly linked to the outbreak's origin without connection between them. Epidemiological data agree with the absence of a link between both Valencian cases, but there exists no evidence of a trip to the Canary Islands for either patient,

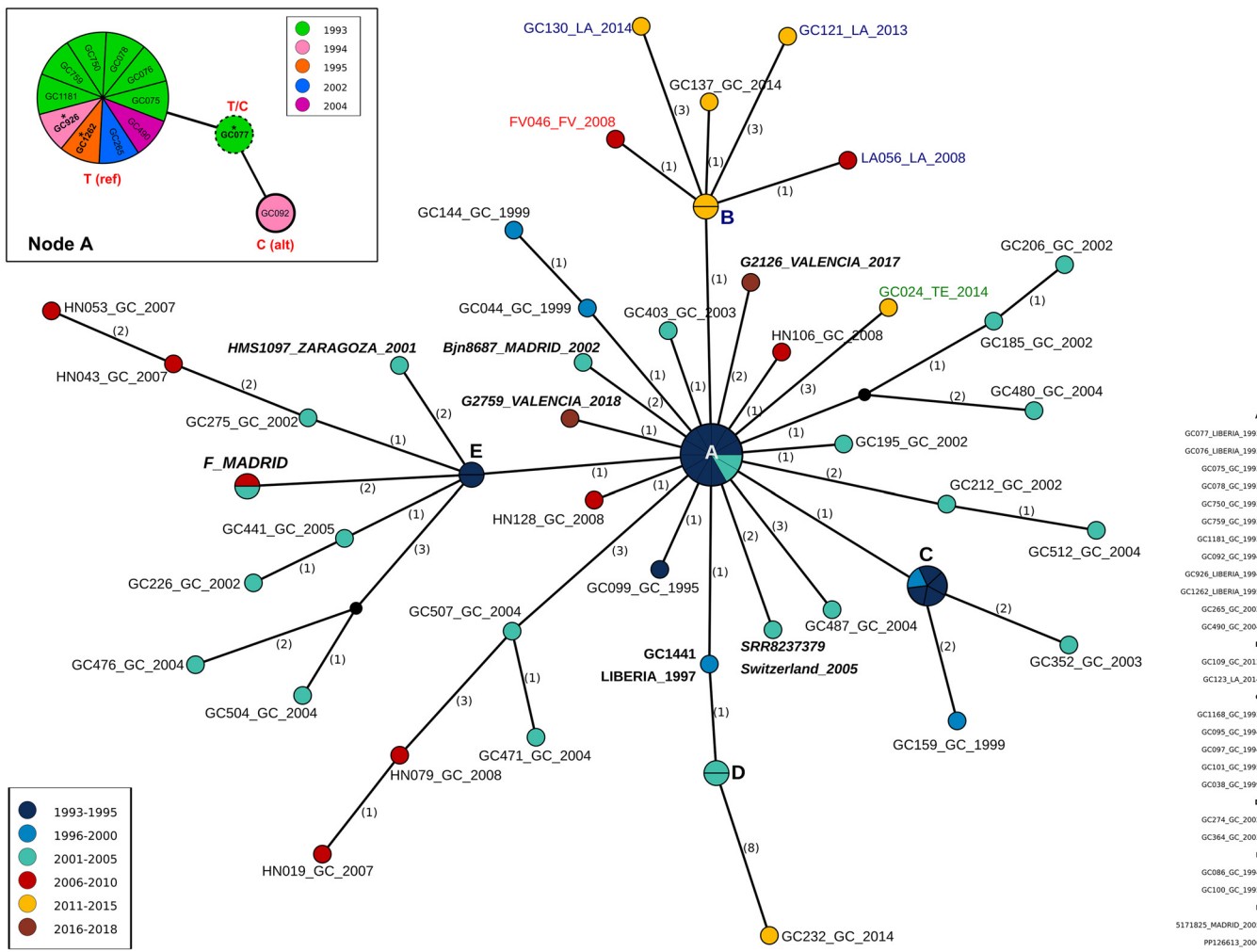

**FIG 3** Median-joining network analysis. Numbers in parentheses indicate the number of SNPs between nodes. Node size indicates the number of samples with the same genome; node color denotes sampling time; name color indicates different islands (blue: Lanzarote; red: Fuerteventura; green: Tenerife; black: Gran Canaria). Samples from the continent are indicated in italics and bold letters. GC1441 is an additional sample from the index case. Node A resolution with hSNP pos: 3910007 is detailed; reference (ref) and alternative (alt) allele distribution among samples is indicated. Names with asterisks indicate samples of the index case.

suggesting an additional missing linking case (or cases). For the 3 isolates collected in Madrid, there exists some link with Gran Canaria; case Bjn8687 was incarcerated in the island 2 years before diagnosis, and the others (PP126613 and 5171825) visited Gran Canaria at some point before diagnosis (Table S1). We did not obtain epidemiological information for the isolate identified from Switzerland. Overall, the extent of secondary transmission after exportation events appears to be severely limited, as we did not retrieve any additional sequences from other locations from the ENA.

**Phylodynamic analysis.** We reasoned that the possession of any transmission advantage by the GC strain, explaining why it caused an outbreak, would be reflected in its natural history. Since we found evidence of temporal structure in our data set (see Materials and Methods), applying a BDSKY model to estimate the epidemiological parameters of the GC outbreak represented a suitable option (Supplementary Notes, Fig. S2 and 3).

The becoming uninfectious or recovery rate ($\delta$) gave a median value of 0.49 (0.28 to 0.75, 95% HPD), suggesting a median infection period of 2 years (1.3 to 3.6 y) in agreement with the previously proposed global estimation of 1 to 3 years (14, 15). The effective reproduction number varies through the study period, displaying a particular profile with peaks of outbreak expansion ($R_e > 1$) and reduction ($R_e < 1$). We observed peaks approximately every 10 years, and with an extension of 3 years (Fig. 4A). Peaks matched with the secondary outbreak nodes observed in the network, and the periods of outbreak reduction coincide with chains instead

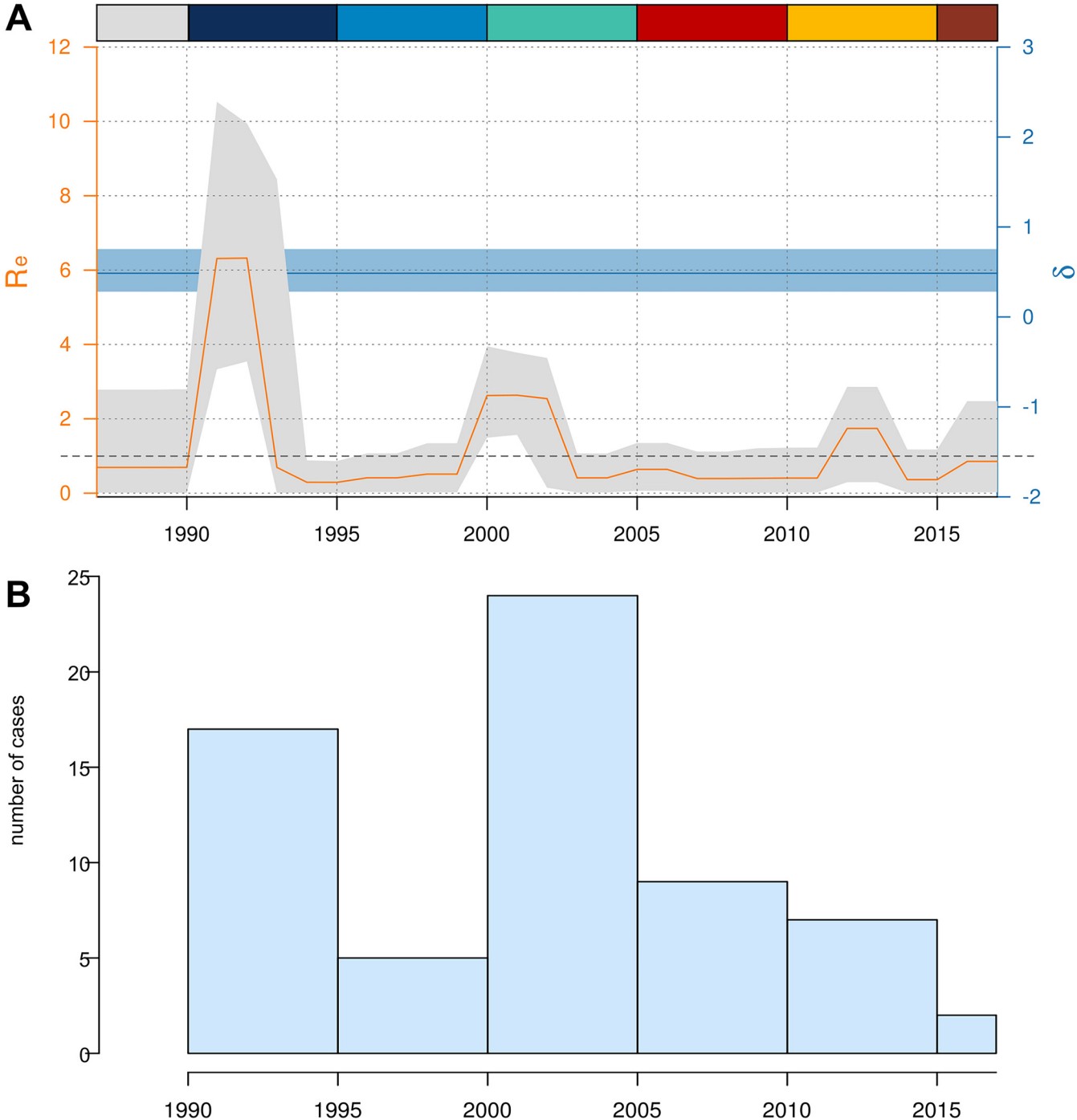

**FIG 4** Phylodynamic analysis of the GC outbreak. (A) BDSKY results showing reproductive number ($R_e$) variation across outbreak time period, and recovery rate ($\delta$). Rectangles in the upper section of the graph indicate periods using the same color scale as nodes in Fig. 3. (B) Histogram of the number of cases sequenced in the different time periods.

of star-like transmission patterns (Fig. 3). As observed in the histogram, peaks do not reflect the sampling effort as 2005 to 2010 and 2010 to 2015 periods display a similar number of samples (Fig. 4B). While isolates appeared in terminal nodes in the first period (Fig. 3), cases were part of a secondary outbreak in the second (Fig. 3, node B). On average, the GC outbreak has decreased since the last period showed a $R_e$ close to 1. Other than its particular profile, we observed a median $R_e$ of 6 (0.01 to 11, 95% HPD), peaking at the beginning of the outbreak (Fig. 4) and far from the value of 10 to 12 as the maximum number of secondary cases caused by an infected person per year reported for TB (14, 16). Of note, ongoing discussions exist

regarding the estimates of infection period and $R_e$ in TB; different factors (e.g., patient immunological status and social determinants) influence both parameters. Nevertheless, our mean values lie within the range of values observed for TB (14, 17), and there is no indication from phylodynamic analysis that the GC strain has a transmission advantage, at least in terms of a higher number of secondary cases generated of lengthened infection period.

If ecological drivers, such as host or social determinants, supported the initial success of the GC strain, infections would be most expected in the vulnerable population. In this sense, 64% of the 61 patients in the GC outbreak presented at least 1 risk factor, including Parenteral Drug Users (PDU), Non-Parenteral Drug Users (NPDU), imprisonment, HIV infection, alcohol abuse, indigence, and a lack of adherence to TB treatment (Table S1). All these factors are well-known risk factors for TB infection in low burden countries. Furthermore, the value increases to 76% when considering only patients with available information. Both values are much higher than the mean value reported for all risk factors during the study period (18).

**Tracking the GC strain over centuries.** The GC strain belongs to MTBC lineage 2. We reconstructed a global ML phylogeny, including 740 L2 global strains (MSA with 43401 concatenated SNPs) from different studies (Table S4). Placed in the context of L2 diversity and considering the previously proposed classification (19, 20), the GC strain belongs to L2.2.3 (Fig. 5A). The closest clade mainly comprises African strains (Fig. 5B, orange clade), followed by a mixed African-Vietnamese clade (Fig. 5B, yellow clade), and a clade more distantly related to China (Fig. 5B). We estimate the emergence of the GC outbreak at around 1984 AD (1973 to 1993 AD, 95% HPD), which agrees with the arrival time of the likely index case to the Canarian archipelago (Fig. S1). We estimate the tMRCA of the African clade as 1930 AD (1913 to 1947 AD, 95% HPD), and the tMRCA of the African-Vietnamese as the middle of the 19th century (Fig. 5B). We estimate the origin of L2.2 as between 559 and 819 AD, which agrees with previous estimates using the same approach (21). Reconstruction of the phylogeographic and dispersal history of the GC strain, places its oldest ancestor in China around the middle of 1300 AD; from there, the GC strain's ancestor spread to Vietnam between 1500 and 1700 AD. From Vietnam, it dispersed to Africa between 1930 and 1940 AD, then to the Canary Islands between 1960 and 1993 AD, before reaching mainland Europe between 1993 and 2002 (Fig. 5C and D, File S1).

## DISCUSSION

*Mycobacterium tuberculosis* is an obligate human pathogen whose success in the population depends on human movement. By combining WGS and phylodynamic analyses, we describe the dynamics of a 25-years outbreak in the Canary Islands in detail. The GC strain quickly spread after the index case arrived in Gran Canaria, as many cases with identical sequences were observed in 1993. A pair of close secondary outbreaks appeared soon after, one connected with the cases in the Canarian archipelago and dispersed to the Spanish peninsula, and another that spread within the same island, probably as an early independent transmission chain. Lately, a differentiated secondary outbreak has spread to the rest of the islands, reaching high representativeness (12). By evaluating serial samples of the likely index case from different years, we identified another late and small secondary outbreak associated with this patient; the epidemiological information linked to this case suggests that poor treatment adherence could explain this patient's most prolonged infectious period and, thus, the generation of that secondary outbreak. A deep analysis of hSNPs resolved the relationships among samples in the central core of the GC outbreak, adding molecular evidence to support GC077 as the index case (first sample). A similar approach has been described by Lee et al. (22), demonstrating the potential of using hSNPs for further genetic discrimination within outbreaks. This approach also provided evidence that most of the observed diversity within the GC outbreak was already present in the index case. Furthermore, this approach allowed us to circumscribe the GC outbreak, discarding distantly related isolates linked by molecular methods that only query a restricted portion of the genome.

With the added value of querying an extensive collection of WGS (13, 23), our comprehensive approach allowed us to link the GC strain to cases in continental Europe and,

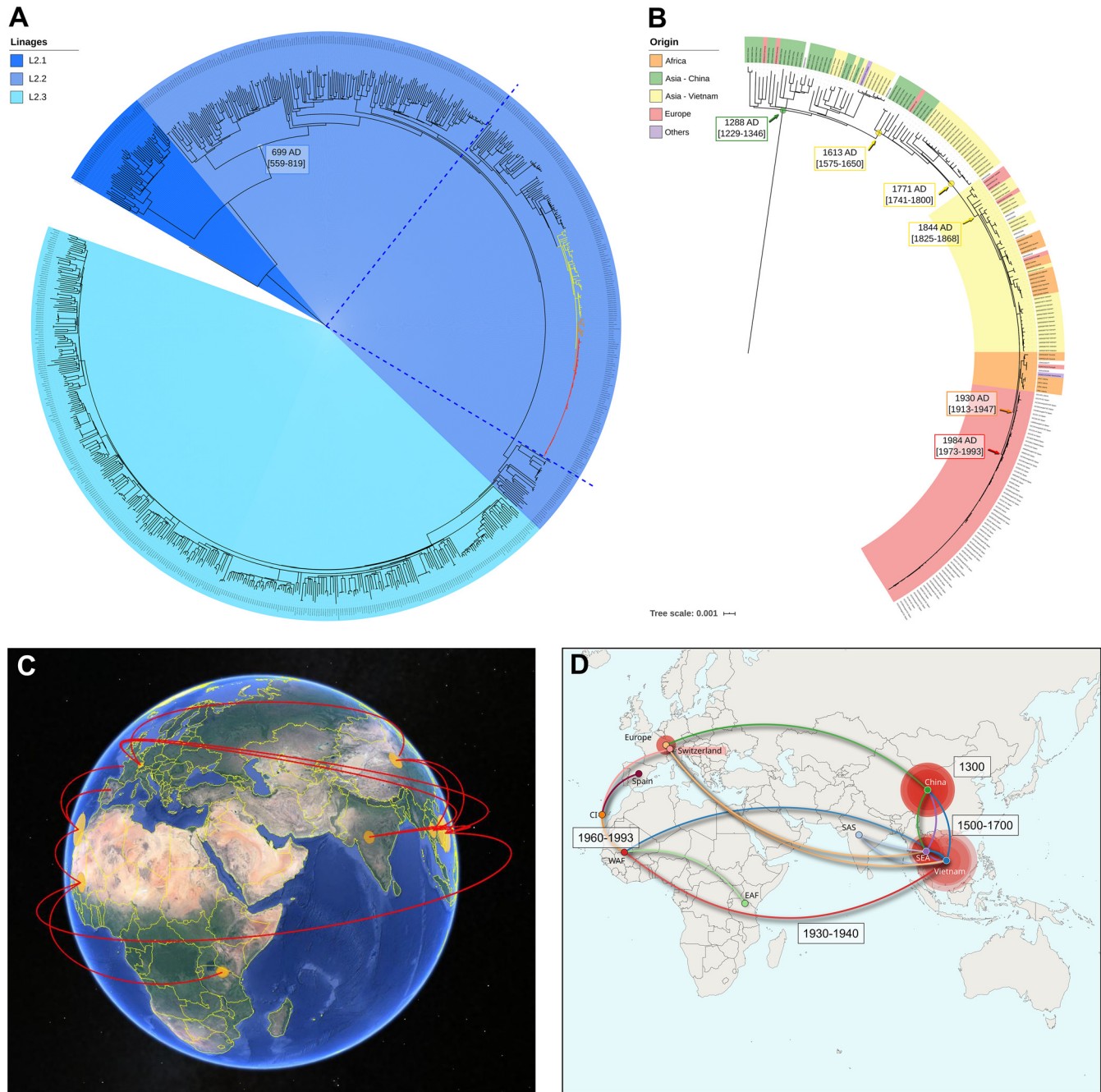

**FIG 5** Dating, phylogeographic, and dispersal analysis. (A) ML global phylogeny of L2 highlighting sublineages. Time of the most recent common ancestor (tMRCA) of L2.2. is indicated. Colored branches correspond to GC strain (red), African-related clade (orange), and African-Vietnamese clade (yellow). (B) tMRCAs and origin of GC strain, closest nodes and oldest Chinese ancestor. Colors indicate the origin of either sample (labels) and the ancestor of different clades. (C and D). The phylogeographic analysis results indicate routes of dispersion since the GC strain's oldest ancestor. (C) 3D view obtained with Google Earth (www.google.com/earth/). (D) Map obtained using the *SPREAD3* software (https://rega.kuleuven.be/cev/ecv/software/SpreaD3_tutorial), lines are colored following the destination of the migration, and times from the oldest origin in China, to Vietnam and the Canary Islands are denoted.

thus, determine its spread beyond the geographic limits of the outbreak. The GC strain had elevated local success, reaching a frequency of 27% of all isolates in Gran Canaria (12) compared to common transmission clusters that account for 1 to 3% of annual cases in low burden settings (24). However, GC strain had limited expansion outside the Canarian archipelago with multiple introductions in continental Europe not resulting in secondary cases (10).

By combining the use of a comprehensive SNP profile and querying a large data set, we shed light on the remote origin of the GC strain. Our analyses place the ancestors of the GC strain hundreds of years ago (around 1300 AD) in China. Thanks to the vast amount of

MTBC genomic data available, we traced the initial movements of GC strain's ancestor from China to Vietnam, Liberia, and, finally, to Gran Canaria by the end of the 20th century. Once in Gran Canaria, the GC strain spread quickly causing an outbreak, with many secondary cases, but with few exportations outside the Canarian archipelago. Thus, we documented how past epidemiological events impact current TB epidemics.

In addition to inferring the remote origin of the GC strain, we had the unique opportunity to study its proximal origin using a phylodynamic approach. Epidemiological information rarely identifies the proximal origins of large outbreaks, mainly due to difficulties in identifying the index patient in an area with many local cases and latency periods (in the case of TB). As the GC outbreak was undoubtedly linked to the migration of an infected person from Liberia to Gran Canaria in 1993 (9), this situation represented a unique opportunity to validate the commonly applied Bayesian approach for TB. Our results (1984 AD [1973 to 1993 AD, 95% HPD]) largely agree with the available epidemiological information.

The low proportion of outbreak cases sequenced from the Canary Islands, which accounts for approximately 10% of total cases, represents the main limitation of our study; however, we demonstrate that the data set has enough phylogenetic signal for the analysis presented. In addition, phylodynamic approaches do not require complete sampling to estimate epidemiological parameters, as evidenced by Stadler et al. and Boskova et al. (25, 26). Our phylodynamic results do not correlate to the sampling effort (Fig. 4). Spain also lacks a national WGS program; however, we used systematic typing data from 3 of the largest regions in Spain in our study (Madrid, Aragón, and the Valencian Region) to confirm that the GC strain displayed a lack of success beyond the Canarian archipelago. By querying the whole ENA database, we retrieved additional cases sequenced elsewhere that could accurately evaluate the expansion of the GC strain. Finally, we performed a similar sequencing effort as in other similar outbreaks; in these situations, the number of cases was sufficient to obtain valuable information on the origin and factors associated with the outbreak (1). Additional limitations at the patient level include the lack of specific information regarding the socio-economic status of all cases and incomplete data regarding the living situation and other risk factors.

A common question regarding MTBC strains is whether some are more transmissible than others or if ecological factors drive their success. In general, outbreak-associated strains represent promising candidates to evaluate if the increasing number of cases with sustained transmission over decades derived from intrinsic strain transmissibility, or other factors like social or host determinants. Outbreak strains have been hypothesized as drivers of local TB burden, but whether they are the cause or consequence of TB transmission remains unclear. For the GC strain, we estimated an infection period close to 2 years and a median of 6 secondary cases per infected individual; both values lie within the ranges expected for TB (14, 16, 17), suggesting that the GC strain does not possess any transmissibility advantage (at least in terms of shorter latency periods or higher contagion rate). In addition, the GC strain did not display higher virulence, accumulated mutations in sequential samples, or mutations associated with resistance (11). Similarly, our extensive query of GC strain related sequences in public repositories did not identify the GC strain as responsible for outbreaks outside the Canary Islands. For example, an in-depth genotyping effort did not identify secondary cases associated with a case in Madrid with prolonged disease (11) or Aragón, where systematic genotyping is applied to all TB positive cases. In addition, mutations associated with the GC strain are unlikely to have a role in transmission. Mutations with a functional role have been proposed for other outbreak-associated strains, notably the Danish C2 (27) and the Toronto (28) strains; however, experimental evidence is unavailable. On the contrary, these outbreak strains share the fact that they thrive in populations with significant risk factors, particularly during the last 20 years of the past century.

In summary, we describe four pieces of evidence supporting a lack of intrinsically higher transmissibility of the GC strain: (i) Previous findings do not suggest any differential biological properties for the GC strain compared to control strains (11); (ii) Epidemiological data support the existence of environmental factors associated with the transmission and, consequently, with the success of the GC strain and the development of the Gran Canaria

outbreak; (iii) There exists little evidence of a similar expansion of the GC strain outside the Canary Islands, as revealed by our global genomic analysis and long-term molecular epidemiology studies in Spain; (iv) Phylodynamic parameters associated with the GC strain lies within the expected range for TB. Overall, our data suggest that the success of the long-lived Gran Canaria outbreak-associated GC strain, and probably others, relates to ecological factors associated with founder effects linked to the host and social determinants of disease.

## MATERIALS AND METHODS

**Study Population.** The epidemiological context of the study setting is described in the Supplementary Notes.

*M. tuberculosis* DNA samples from 80 different patients plus 3 additional samples from the index case, all from Canary Islands, and 1 from Zaragoza were supplied by the Instituto de Investigación Sanitaria de Aragón (Spain). In addition, DNA samples from 5 patients from the Spanish peninsula (Madrid) were supplied by the Hospital Universitario Gregorio Marañón, 2 of which belonged to recent immigrants from Guinea (Table S1). A representative number of samples were selected from the Canary Islands (i.e., at least 15 samples every 5 years during the study period 1993 to 2014), to ensure a balanced representation of the diversity of the GC strain. All samples were previously identified as part of the Gran Canaria outbreak by genotyping methods.

Ethics approval was obtained from the Ethics Committee, Hospital Universitario de Gran Canaria Dr. Negrín, Las Palmas de Gran Canaria (Code 2019-502-1) and by the Ethics Committee for Clinical Research from the Valencia Regional Public Health Agency (Comité Ético de Investigación Clínica de la Dirección General de Salud Pública y Centro Superior de Investigación en Salud Pública). Informed consent was waived as TB detection forms part of the compulsory regional surveillance program for communicable diseases. All personal information was anonymized, and no data allowing patient identification was retained. The epidemiological information was obtained from Servicio de Microbiología, Hospital Universitario de Gran Canaria Dr. Negrín under the Ethics Committee Application: CEIm H.U.G.C. Dr. Negrín 2019-502-1.

**Study design.** The whole genome of *M. tuberculosis* from the first set of 89 samples (86 patients) was sequenced. Additional samples were searched for by querying our local database of sequences (23, 24, 29), with the 3 previously selected Gran Canaria outbreak-specific SNPs or partial SNP profile (10); Rv2524 (C1398T and SNP2847935), Rv3869 (G1347C and SNP4346385), and Rv0926c (G162A and SNP1033625). The redefinition of specific SNPs, or complete SNP profile, was conducted based on phylogeny and distance analysis. New samples were identified by inspecting the whole ENA repository and samples from ongoing projects. Phylogeographic and phylodynamic analyses were performed in order to evaluate the outbreak. Comparisons with other outbreaks were also performed; sequences corresponding to the Denmark (PRJEB20214) (1), Thailand (PRJNA244659) (4), Bern (PRJEB5925) (6); and Buenos Aires ([BA], PRJEB7669) (5) outbreaks were downloaded from the ENA and analyzed with the bioinformatics pipeline detailed below.

**Whole-genome sequencing and bioinformatics analysis.** DNA samples were used to prepare sequencing libraries with a Nextera XT DNA library preparation kit (Illumina), following the manufacturer's instructions. Sequencing was performed on an Illumina MiSeq instrument, applying a $2 \times 300$ bp paired-end chemistry. The sequencing was performed at the Instituto de Biomedicina de Valencia (Spain). The general bioinformatics analysis is described at https://gitlab.com/tbgenomicsunit/ThePipeline/-/tree/master/. Briefly, read files were trimmed and filtered with fastp (30). Non-MTBC reads were removed with Kraken software (31). Sequences were then mapped to an inferred MTBC common ancestor genome (https://doi.org/10.5281/zenodo.3497110) using bwa (32). SNPs were called with SAMtools (33) and VarScan2 (34). GATK HaplotypeCaller (35) was used for InDel calling. SNPs with a minimum of 20 reads (20X) in both strands and quality 20 were selected. InDels with less than 20X were discarded. SnpEff was used for SNP annotation using the H37Rv annotation reference (NC_000962). Finally, SNPs falling in genes annotated as PE/PPE/PGRS, 'maturase', 'phage', '13E12 repeat family protein', those located in insertion sequences, and those within InDels or in higher density regions (>3 SNPs in 10 bp) were removed, due to the uncertainty of mapping. Heterogenous SNPs (hSNPs) were obtained from filtered SNP files; they were classified as positions where >5% and <90% of the reads were the alternative allele (22); hSNPs were searched only for those positions for which at least 1 sequence has that SNP as fixed (i.e were >90% of reads were the alternative allele). Lineages were determined by comparing called SNPs with specific phylogenetic positions established (36, 37). An in-house R script was used to detect mixed infections based on the frequency of lineage- and sublineage-specific positions (resistance_phylogeny_coinfection_v4.R, https://gitlab.com/tbgenomicsunit/ThePipeline/-/tree/master/tools) (29). Pairwise distances were computed with the R *ape* package, and statistical analysis and graphics were performed with R. Raw sequencing data are available under the accession number PRJEB50491 (ENA).

All WGS Illumina sequencing runs stored in the ENA with metadata identifying them as MTBC as of July 2018 were downloaded (N = 38075). A de Bruijn graph (k = 31) was built from each with Cortex v1.0.5.21 (https://github.com/iqbal-lab/cortex/releases/tag/v1.0.5.21), and sequencing errors were removed by excluding low coverage unitigs from the graph (threshold is sample dependent and automatically chosen by the software). The remaining kmers from these graphs were then used to build a BItsliced Genomic Signature Index (BIGSI) (13). The index was used to query these ENA samples for the outbreak-related SNPs (partial and complete SNP profiles), by first creating two 61 base-pair probes for each SNP (30 bp flanking each side of the SNP, from the reference genome, and 1 probe for each SNP allele), returning binary information as to which sequencing runs contained all the kmers in the probes.

**Phylogenetic analysis.** Multisequence alignment (MSA) files were constructed with concatenated SNPs discarding well-known drug resistance and invariant positions. Maximum likelihood trees were inferred with RAxML v8.2.11 (38) using the GTRCATI model and 1000 fast-bootstrap replicates. Tree visualization and editing were conducted in ITOL (https://itol.embl.de/). Specific SNPs, or complete SNP profile, were identified using likelihood ancestral reconstruction of Mesquite software v3.61 (http://www.mesquiteproject .org). The genomic network was constructed with the MSA files with a median-joining network inference method implemented in PopArt Software (39).

**Time and geographic origin of the GC strain.** A set of 200 samples including those from the Gran Canaria outbreak, closest clades, and a representative subset of global samples selected with Treemer (40), was used to estimate the time of the most common ancestor (tMRCA) of the GC strain and deeper nodes. The evolutionary history of the GC strain was reconstructed with BEAST2 v2.5.1 (41) using GTR (gamma 4) as site model, coalescent constant population as tree prior, and relaxed clock log-normal with gamma distribution as prior. A tip dating method was used and the clock rate set based on previous publications (mean = $4.6 \times 10^{-8}$ substitutions per site per year with 95% HPD: $3.3 \times 10^{-8}$ to $6.2 \times 10^{-8}$) (42). Ascertainment bias was corrected by adjusting the clock rate based on the alignment size (24). Three independent runs of Markov Chain Monte Carlo (MCMC) length chains of 10 million, sampling every 1000 steps, were performed to reach convergence. Log and tree files were combined with the LogCombiner tool, discarding 10% of burn-in. Combined files were inspected with Tracer v1.7.1 (43). All parameters reached effective sample size (ESS) > 200 and well mixing. The tree was annotated with TreeAnnotator and visualized with FigTree v1.4.3.

Spatial dispersal dynamics of the GC strain and closest clades were reconstructed using a phylogeographic diffusion in discrete space approach implemented in BEAST v2.5.1 (Beast-classic package) (44). A set of 142 samples, including the whole outbreak and closest clades, and those global strains with available information of year and location were used. The same models, parameters, as in dating analysis, were used. The SPREAD3 tool was used for geographic visualization of GC strain evolutionary history (45), only dispersion patterns with posterior values higher than 0.9 were plotted. The maps in Fig. 5 were created as detailed in the SPREAD3 tutorial (https://rega.kuleuven.be/cev/ecv/software/SpreaD3 _tutorial), and also visualized with Google Earth (www.google.com/earth/). The map in Fig. 2 was constructed using the R package *ggspatial* (https://r-spatial.org/r/2018/10/25/ggplot2-sf-2.html).

**Outbreak phylodynamics.** Outbreak dynamics were studied with the Bayesian Birth-Death Skyline (BDSKY) model implemented in BEAST2 v2.5.1. The BDSKY model estimates parameters related to outbreak's evolution (25). The analysis was carried out with all samples included in the Gran Canaria outbreak (64 samples) (Table S1). We use a GTR (gamma 4) as site model, and a strict clock with LogNormal prior distribution, and the mean value previously published (42). The 'Birth-death skyline serial' tree model was used. Specific parameters were set as follows: (i) becoming uninfectious (LogNormal; M = 0, S = 1.25), considering the global estimation of 1 year as TB infectiousness period and not considering latency (14); (ii) origin (LogNormal; M = 30; S = 0.04) considering the likely index case arrived in the archipelago at the beginning of 1993, and initiating the outbreak immediately; (iii) reproductive number (LogNormal; M = 0; S = 1.25; dimension = 13) considering that each person can generate as maximum 12 secondary cases in settings with low TB burden (14); and (iv) sampling proportion (beta; alpha = 3; beta = 50), calculated as the total number of TB cases in the Canary Islands between 1993 and 2017 (https://www3.gobiernodecanarias.org/sanidad/scs/listaImagenes.jsp?idDocument= fe9e3e94-feee-11e0-ab85-376c664a882a&idCarpeta=0f67aaf7-9d88-11e0-b0dc-e55e53ccc42c), and considering that 20 to 30% of cases belonged to the outbreak, resulting in a sampling range of 8 to 13%. MCMC's chain length of 10M was run with sampling every 1000 steps for the posterior distribution. The log file was inspected with Tracer v1.7.1 (43), all parameters reached ESS > 200 and well mixing. Results were inspected and plotted with the R package *bdskytools* (https://github.com/laduplessis/bdskytools).

The clocklike structure of the data set was first evaluated with a linear regression analysis of tip dates vs root-to-tip distances with TempEst (46), and by applying a date randomization test (DRT) with 100 randomized replicates of the BDSKY analysis. The clocklike structure was evaluated with the analysis proposed by Menardo et al. (47), considering that DRT is passed when the clock rate estimate for the observed data does not overlap with the range of estimates obtained from the randomized sets, intermediate DRT is passed when the clock rate estimate for the observed data does not overlap with the confidence intervals of the estimates obtained from the randomized sets, and stringent DRT is passed when the confidence interval of the clock rate estimate for the observed data does not overlap with the confidence intervals of the estimates obtained from the randomized sets.

**Data and code availability.** All sequences have been deposited in the ENA repository under project number PRJEB50491. All scripts, tools, and reference sequences are included in the repository section of our laboratory webpage (http://tgu.ibv.csic.es/?page_id=1794) and GitLab page (https://gitlab.com/tbgenomicsunit/ ThePipeline). Project numbers for the additional sequences are listed in Tables S2 and S4. Any additional information required to reanalyze the data reported in this paper will be made readily available from the lead contact upon request.

## SUPPLEMENTAL MATERIAL

Supplemental material is available online only.

**SUPPLEMENTAL FILE 1**, PDF file, 0.8 MB.

**SUPPLEMENTAL FILE 2**, XLSX file, 0.1 MB.

**SUPPLEMENTAL FILE 3**, AVI file, 2.6 MB.

## ACKNOWLEDGMENTS

This project has been funded by the European Research Council (101001038-TB-RECONNECT), the Ministerio de Economía, Industria y Competitividad (PID2019-104477RB-I00), and the European Commission–NextGenerationEU (Regulation EU 2020/2094), through CSIC's Global Health Platform (PTI Salud Global) to I.C. This project has been funded by the Instituto de Salud Carlos III (FIS18/0336) and the Gobierno de Aragón/Fondo Social Europeo "Construyendo Europa desde Aragón" to SS.

I.C. received consultancy fees from Foundation for Innovative New Diagnostics (FIND). The author has no other competing interests to declare. The remaining authors declare no competing interests.

M.G.L. and I.C. conceived the study; M.I.C.H., F.C., R.C., L.S., B.P., L.P.L., D.G.V., S.S., J.C., and E.L. collected the samples, obtained, and curated the epidemiological data; M.T.P. processed and sequenced the samples; P.W. and Z.I. performed BIGSI analysis; M.M.M. performed the scripts for DST analysis; M.G.L. performed all analyses; M.G.L. and I.C. wrote the first draft of the manuscript, with revisions from all coauthors.

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
