## [Reviewer comments · Microbiology Spectrum]

Microbiology Spectrum

Deciphering the tangible spatio-temporal spread of a 25-years tuberculosis outbreak boosted by social determinants

Mariana Lopez, María Campos-Herrero, Manuela Torres-Puente, Fernando Cañas, Jessica Comín, Rodolfo Copado, Penelope Wintringer, Zamin Iqbal, Eduardo Lagarejos, Miguel Moreno-Molina, Laura Pérez-Lago, Berta Pino, Laura Sante, Darío García de Viedma, Sofía Samper, and Iñaki Comas

Corresponding Author(s): Mariana Lopez, Instituto de Biomedicina de Valencia

Review Timeline:

Submission Date:	August 8, 2022
Editorial Decision:	September 3, 2022
Revision Received:	November 1, 2022
Editorial Decision:	November 10, 2022
Revision Received:	January 13, 2023
Editorial Decision:	January 16, 2023
Revision Received:	January 17, 2023
Accepted:	January 18, 2023

Editor: Shannon Manning

Reviewer(s): The reviewers have opted to remain anonymous.

Transaction Report:

DOI: <https://doi.org/10.1128/spectrum.02826-22>

September 3, 2022

Dr. Mariana G Lopez
Instituto de Biomedicina de Valencia
Valencia
Spain

Re: Spectrum02826-22 (**Deciphering the tangible spatio-temporal spread of a 25 years tuberculosis outbreak boosted by social determinants**)

Dear Dr. Mariana G Lopez:

Thank you for submitting your manuscript to Microbiology Spectrum. We have received reports from 2 reviewers and would like to invite you to revise your work for further consideration. Indeed, the reviewers and I agree that this manuscript represents a sound analysis of an important pathogen. We also agree that it could be revised for English grammar while paying close attention to syntax. We also agree that additional information is needed about the TB epidemic to bolster the findings and discussion. Please consider these points in your revision.

Link Not Available

Sincerely,

Shannon Manning

Journals Department
Reviewer comments:

Reviewer #1 (Comments for the Author):

This is a well done study with a well written manuscript, showing through analysis beyond an outbreak that a successful M. tuberculosis strain is not necessarily inherently better. It could be successful in one place, due to human/social factors and have an unremarkable trajectory elsewhere.

Two suggestions:

The authors describe an outbreak and they describe a strain.

They use the classical definition of outbreak in the introduction and this is good. This means that the word outbreak, throughout the paper, should be restricted to the description of events on the islands.

When the bacteria travels elsewhere, in space and time, it would avoid confusion if they simply called it a strain (GC strain), as they have often done. There are times that they refer to a phylogenetic outbreak (line 305), but the phylogeny is an analysis of bacteria, not of an epidemiological event. On line 239, they describe the method of visualizing an outbreak evolutionary history. Is this not the evolutionary history of the GC strain? Figure 2 legend mentions genomic delineation of the outbreak, but this is a map of the world. Should this genomic delineation of the GC strain? Same in the legend, 2D. This is how it presented on line 406.

Second point: The major finding is that ordinary strains can look remarkable if they are introduced into a high transmission environment. I agree with this. Yet, in the introduction, line 117, the authors write that outbreak strains are usually drivers of local TB burden. We don't know if they are drivers or passengers, do we? The authors can set up their paper better by saying that these outbreak strains have been associated with local TB burden or linked to local TB burden, but whether they are the cause or consequence of TB transmission is not always clear. Or they can say that outbreak strains have been hypothesized to be the drivers of local TB burden. This way they can point out that it is commonly written that the strain is responsible for the epidemiology, but that this widely held view merits greater scrutiny.

Reviewer #3 (Comments for the Author):

López et al. present an in-depth investigation into a 25 year TB outbreak in the Canary Islands. The methodology is sound and the details surrounding WGS and the analysis are well presented. The finding of host-related factors contributing to transmission are notable, and this assessment will be of interest to the TB community as well as a case study of outbreak investigations in a unique setting. A few comments are below, notably the need for additional context to TB transmission and prevalence in the Canary Islands in the introduction and methods:

1. I am not familiar with expected prevalence of strains by transmissibility factors, or the relative proportion of cases that can be expected from a single outbreak- how does prevalence in 20% of circulating strains compare to typical outbreaks after such a long period of time? Why did the authors hypothesize that this outbreak was good for a transmissibility study and not rather a more typical variation? Additional context would be informative to this regard, especially given the note in the discussion that the proportion of sequenced cases represents only 10% of the total.
2. The authors state in Ln 423 "Overall, there is no indication from phylodynamics analysis that the GC strain has a transmission advantage," but again some context would be helpful to clarify expected results and thresholds to evaluate high transmissibility as well as to give some additional background into the occurrence of TB and expected rates in the Canary Islands given historical studies, especially in different populations with various risk factors for disease.
3. The authors helpfully provide reasons and hypotheses for their findings for the GC strain outbreak, though the rationale is often missing. For example, Ln 485- why is it thought transmission was due to poor treatment adherence, rather than lack of diagnosis or other reasons? Were treatment-related mutations identified in the secondary strains? Specific statements should be supported with study or historical data especially to suggest effective TB control measures.
4. Additional limitations in terms of the patient-level data available for full evaluation of the transmissibility of the strains should be mentioned (SES, living situation, etc.).

Minor comments

1. The submission should be revised for English grammar, as there are many syntactical errors, especially in the Introduction and discussion
2. Please clarify Ln 148: "A representative number of samples from the Canary Islands"
3. Where was WGS performed? In particular, please describe the infrastructure and sample referral/flow processes in the Canary Islands that contribute to the success of such an outbreak investigation.
4. Consider providing the in-house R script in supplement
5. Figure 5C: please adjust coloring so that locations and lines are clearer

Staff Comments:

Preparing Revision Guidelines

Please return the manuscript within 60 days; if you cannot complete the modification within this time period, please contact me. If you do not wish to modify the manuscript and prefer to submit it to another journal, please notify me of your decision immediately so that the manuscript may be formally withdrawn from consideration by Microbiology Spectrum.

Dear Editor,

We want to thank all reviewers and editors for their valuable comments and suggestions which significantly improved our work. We addressed all issues raised and have marked the changes in the file named as "Lopez_et_al_GCoutbreak_marked.pdf"

Reviewer #1 (Comments for the Author):

This is a well done study with a well written manuscript, showing through analysis beyond an outbreak that a successful M. tuberculosis strain is not necessarily inherently better. It could be successful in one place, due to human/social factors and have an unremarkable trajectory elsewhere.

Two suggestions:

The authors describe an outbreak and they describe a strain.

They use the classical definition of outbreak in the introduction and this is good. This means that the word outbreak, throughout the paper, should be restricted to the description of events on the islands.

When the bacteria travels elsewhere, in space and time, it would avoid confusion if they simply called it a strain (GC strain), as they have often done. There are times that they refer to a phylogenetic outbreak (line 305), but the phylogeny is an analysis of bacteria, not of an epidemiological event. On line 239, they describe the method of visualizing an outbreak evolutionary history. Is this not the evolutionary history of the GC strain? Figure 2 legend mentions genomic delineation of the outbreak, but this is a map of the world. Should this genomic delineation of the GC strain? Same in the legend, 2D. This is how it presented on line 406.

We want to thank the reviewer for their advice in clarifying the use of the terms outbreak and strain. We revised the manuscript and considered the outbreak as the clade defined in the phylogeny (Fig 2A), including all the samples from the Canary Islands, two from Valencia, and one from Switzerland. All these samples are genomically related, and also related in time and space, even when not wholly restricted to the Canarian archipelago. In addition, most sequences from the Spanish peninsula have known epidemiological links with the islands. Outbreaks can spread behind the area of origin, as we demonstrate in our study. Following the reviewer's recommendations, we differentiate the Gran Canaria outbreak (as defined by the differentiated clade in the phylogeny) from the GC strain, with the latter term used to describe the phylogeographic results.

On line 234 (former 239), we modified "outbreak evolutionary history" by "GC strain evolutionary history"

On line 297 (former 305), by "phylogenetic definition of the outbreak" we mean the method used to define the outbreak. We reconstructed a phylogeny using all samples previously assigned to the outbreak by molecular typing methods and all the strains harboring the outbreak marker SNPs defined by Perez-Lago et al. (2016). The clade identified as "the outbreak" includes all samples from the Canary Islands and a few closely related in time and space and very likely genomically and epidemiologically linked. Thus, the samples (even from outside the Canarian archipelago) are considered part of the outbreak. We added a

sentence to clarify the definition of the outbreak by using a phylogenetic approach based on whole genome sequencing data.

Line 297: “This clade likely represents the phylogenetic definition of the outbreak based on WGS data and is referred to as “outbreak” hereafter. All samples included are genomically linked, and most cases are geographically and epidemiologically linked.”

Following the reviewer’s suggestion, we have also modified the legend of Figure 2 as follows:

“**Figure 2.** Outbreak characterization and sequence distribution **A.** ML tree highlights the outbreak’s phylogenetic circumscription and related strains identified with the partial SNP profile. **B-C.** Density graphs of the pairwise number of SNPs between samples of different outbreaks Gran canaria (GC); Buenos Aires (BA) (5); Bern (6); Denmark (4); Thailand (3). **D.** Distribution of sequences belonging to the outbreak (squares) and related to GC strain (circles). Colors indicate the source and shapes denote the meeting SNP profile of each sequence included in the study. The map was obtained from the R package *rnaturalearth* (<https://docs.ropensci.org/rnaturalearth/articles/rnaturalearth.html>)”

Second point: The major finding is that ordinary strains can look remarkable if they are introduced into a high transmission environment. I agree with this. Yet, in the introduction, line 117, the authors write that outbreak strains are usually drivers of local TB burden. We don't know if they are drivers or passengers, do we? The authors can set up their paper better by saying that these outbreak strains have been associated with local TB burden or linked to local TB burden, but whether they are the cause or consequence of TB transmission is not always clear. Or they can say that outbreak strains have been hypothesized to be the drivers of local TB burden. This way they can point out that it is commonly written that the strain is responsible for the epidemiology, but that this widely held view merits greater scrutiny.

We thank the reviewer for their comments, which have improved our manuscript. We modified the introduction changing “... drivers of local TB burden” to “... associated with local TB burden”. In addition, we include the following sentence in the discussion in agreement with the reviewer's suggestion:

Line 520: “Outbreak strains have been hypothesized as drivers of local TB burden, but whether they are the cause or consequence of TB transmission remains unclear.”

Reviewer #3 (Comments for the Author):

López et al. present an in-depth investigation into a 25 year TB outbreak in the Canary Islands. The methodology is sound and the details surrounding WGS and the analysis are well presented. The finding of host-related factors contributing to transmission are notable, and this assessment will be of interest to the TB community as well as a case study of outbreak investigations in a unique setting. A few comments are below, notably the need for additional context to TB transmission and prevalence in the Canary Islands in the introduction and methods:

1. I am not familiar with expected prevalence of strains by transmissibility factors, or the relative proportion of cases that can be expected from a single outbreak- how does prevalence in 20% of circulating strains compare to typical outbreaks after such a long period of time? Why did the authors hypothesize that this outbreak was good for a transmissibility study and not rather a more typical variation? Additional context would be informative to this regard, especially given the note in the discussion that the proportion of sequenced cases represents only 10% of the total.

We thank the reviewer for raising this point. Regarding the outbreak, it is crucial first to discuss the expected cluster sizes in a TB setting. Cluster sizes in TB using genomic data, and using the same definition as our manuscript, typically range from 2 to 10 cases in low-burden countries (see Walker et al. 2014), considering a population-based study for six years and 384 cases. The size of the clusters depends on the timespan and other factors, but they usually represent a low percentage of total annual cases. In reality, TB epidemics in a low-burden country are composed of multiple small clusters originating from genetically different genotypes. For context, we provide data from other regions of Spain (Valencia) obtained in a population-based study of three years, where cluster size ranges between 2 and 12 (median of 2) (Cancino-Muñoz et al., 2022). Thus, the annual prevalence of any of these genotypes (considering annual TB incidence in Valencia is around 400 cases) is around 1% for the largest cluster genotype. By contrast, the GC strain represents ~20% of prevalent annual cases in the Canary Islands (Pérez-Lago et al., 2019), a much greater contribution than a typical TB genotype. In addition, the GC was declared an outbreak as it was identified as a large cluster, accounting for 75 cases in only four years (Caminero et al., 2001), a significantly higher number of cases than common transmission clusters. Furthermore, Folkvardsen et al. (2017) reported a TB outbreak in Denmark, circulating for 23 years, which accounted for 18% of the total cases in transmission, similar to our observation. Due to the increased number of cases within outbreaks (renamed outbreaks to differentiate from ordinary transmission clusters), it has always been considered that strains causing outbreaks have high intrinsic transmissibility; however, this has never really been tested (which is what the reviewer enquires in Question 2, see answer below).

These extraordinary outbreaks make them candidates in the search for intrinsically highly transmissible strains (which represents the common belief) versus other factors like social or host determinants of disease. In this case, we use the GC strain as an example of an outbreak-associated strain that significantly contributes to the local TB burden.

Regarding the proportion of cases, we performed sequencing (as in other similar outbreaks, Folkvardsen et al., 2017), in which the number of cases was sufficient to obtain valuable information regarding the origin and factors associated with the outbreak. Importantly, phylodynamic approaches do not need complete sampling to estimate epidemiological parameters. This has been recently illustrated in studies of SARS-CoV-2 variants where factors like R_e have been robustly estimated from partially sampled data. In addition, we do not rely only on phylodynamic analysis in our case. We made a comprehensive effort to identify recent transmission cases deposited in databases and searched the most extensive public repository for additional sequences associated with the outbreak, harboring the defined SNP profiles to include more cases if available.

We now mention the Valencia data and data from other countries to provide some context to the relevance of GC strains and other outbreak strains. In the introduction, we indicate:

Lines 109 "...raising the question of whether these strains have any intrinsic transmissibility advantage or if their success derived from local population processes such as founder effects or ecological drivers of transmission (3–8)."

We have added the following paragraphs to the discussion

Line 479 : "The GC strain had elevated local success, reaching a frequency of 27% of all isolates in Gran Canaria (12) compared to common transmission clusters that account for 1-3% of annual cases in low-burden settings (46). However, GC strain had limited expansion outside the Canarian archipelago with multiple introductions in continental Europe not resulting in secondary cases (10)."

Line 500: "The low proportion of outbreak cases sequenced from the Canary Islands, which accounts for approximately 10% of total cases, represents the main limitation of our study; however, we demonstrate that the dataset has enough phylogenetic signal for the analysis presented. In addition, phylodynamic approaches do not require complete sampling to estimate epidemiological parameters, as evidenced by Stadler et al. and Boskova et al. (34, 47). Our phylodynamic results do not correlate to the sampling effort (Figure 4). Spain also lacks a national WGS program; however, we used systematic typing data from 3 of the largest regions in Spain in our study (Madrid, Aragón, and the Valencian Region) to confirm that the strain displayed a lack of success beyond the Canarian archipelago. By querying the whole ENA database, we retrieved additional cases sequenced elsewhere that could accurately evaluate the expansion of the outbreak. Finally, we performed a similar sequencing effort as in other similar outbreaks; in these situations, the number of cases was sufficient to obtain valuable information on the origin and factors associated with the outbreak (4)"

Line 517: "In general, outbreak strains represent promising candidates to evaluate if the increasing number of cases with sustained transmission over decades derives from intrinsic strain transmissibility or other factors like social or host determinants."

2. The authors state in In 423 "Overall, there is no indication from phylodynamics analysis that the GC strain has a transmission advantage," but again some context would be helpful to clarify expected results and thresholds to evaluate high transmissibility as well as to give some additional background into the occurrence of TB and expected rates in the Canary Islands given historical studies, especially in different populations with various risk factors for disease.

We use the phylodynamic analysis to infer the R_e value for the outbreak. Estimates of R_e for TB are diverse, as R_e depends on intrinsic transmissibility of the bacteria and the environmental context (e.g., susceptible population, number of contacts, etc). We do not have strict values for both parameters, and much discussion exists around them, as discussed by Dowdy et al. (2013) and Ma et al. (2018); distinct factors, such as patient immunological status or social determinants, influence both parameters. However, our overall values lie within the range observed for TB. A recent metanalysis suggests an R_e

range of around 0.25-4 (Ma et al., 2018), while other studies report values between 10-12 (Dowdy et al., 2013; Pai et al., 2016). The GC strain peaked at 6 during the first years of expansion, as expected given the prevalence of high-risk individuals in the Canary Islands (see below the epidemiological context) but has remained well below 3 after the initial expansion.

Our rationale was that higher intrinsic transmissibility of the GC strain (for example through prolonged infection period or a shortened latency period), will lead to more secondary cases and an increased R_e than in epidemiological settings. In our analysis, both the estimates of R_e and the infectious period generally lie within the ranges expected and thus do not suggest an increased intrinsic transmissibility of the GC strain. In addition, we do not rely solely on phylodynamic analysis. There exist four pieces of evidence that we believe support the conclusion that the GC strain does not have intrinsically higher transmissibility: i) In agreement with our results, previous findings suggest no different biological properties for the GC strain (Pérez-Lago et al., 2015) compared to control strains; ii) Our epidemiological data support the association of environmental factors with the transmission and, consequently, with the GC strain's success and the outbreak development; iii). There exists little evidence of a similar expansion of the GC strain outside the Canary islands, as revealed by our global genomic analysis and by long-term molecular epidemiology studies in Spain; and iv) As noted, the phylodynamic parameters associated with the GC strain lie within the expected range for TB. Therefore we do not rely solely on phylodynamic data for our statement.

Regarding the epidemiological context of TB in the Canary Islands, many records come from the most populated island (Gran Canaria) (Caminero et al.; 1991; Caminero et al., 1995; Caminero et al., 1999), while one report studying the GC strain also mentioning the other islands (Pérez-Lago et al., 2019). Before the outbreak onset, the incidence of TB in Gran Canaria declined from 32.2 cases/100,000 in 1988 to 29.4 in 1992 (Caminero et al., 1995, see Fig 1). During those years, alcoholism was the higher risk associated with TB, accounting for ~35% of cases; the other risk factors were present in less than 10% of cases (Caminero et al., 1995. Table 1). We observed a decreasing trend in diagnostic delay, with most cases diagnosed before one month. During the first outbreak years (1993-1999), TB incidence suffered a slightly increasing trend from 28.5 (1993) to 29.0 (1996) and 29.3 (1999) (Caminero et al., 2001). The only risk factor reported as associated with TB was HIV infection (14% of cases) (Caminero et al., 1999).

More records have been available since 2000 when the TB surveillance program from the Canarian Public Health Service started; the incidence followed a constant decrease in all islands (Figure 2). The risk factors associated with TB existed in between 3-20% of cases, with the majority of risk factors only observed in 10% of cases (Table 2). In our analysis, 64% of cases had at least one TB-associated risk factor; this value is higher than the reported population values for the studied period, which agrees with our conclusion that the success of the GC strain associates with host or environmental factors than with intrinsic strain transmissibility.

Figure 1. Trends in TB in Gran Canaria from 1988-92. Incidence rates per 100,000 population. From Caminero et al., 1995.

Data	1988	1989	1990	1991	1992
Total cases	214	195	174	188	196
Males					
No. of cases	149	136	118	130	129
Rate/100 000	45.6	41.6	36.1	39.2	38.9
Females					
No. of cases	65	59	56	58	67
Rate/100 000	19.9	18.0	17.1	17.3	20.0
High risk factors*					
HIV infection	7.6%	6.4%	7.2%	9.6%	9.6%
Drug abuse (i.v.)	5.5%	6.4%	8.5%	6.2%	6.2%
Malnutrition	7.6%	5.8%	4.6%	10.3%	9.1%
Alcoholism	35%	33.7%	36.8%	36.8%	34.6%
Diabetes	7.1%	4.0%	6.6%	3.4%	4.5%
Other	5.3%	3.4%	6.7%	4.2%	5.7%
Diagnostic delay					
< 1 month	76 (35.5%)	93 (47.7%)	74 (42.5%)	81 (43.1%)	103 (52.6%)
1-3 months	52 (24.3%)	36 (18.5%)	43 (24.7%)	67 (35.6%)	50 (25.5%)
> 3 months	86 (40.2%)	66 (33.9%)	57 (32.8%)	40 (21.3%)	43 (21.9%)
Cured patients	45 (21%)	126 (64.6%)	133 (76.4%)	160 (85.1%)	168 (85.7%)
Abandonment of treatment [†]	157 (73.4%)	63 (32.3%)	38 (21.8%)	13 (6.9%)	18 (9.2%)
Mortality	12 (5.6%)	6 (3.1%)	3 (1.7%)	15 (8%)	10 (5.1%)

*Percentage of the total number of cases for which these data were available; [†]Patients failing to attend follow-up visits are included.

Table 1. Tuberculosis risk factors in Gran Canaria during 1988-1992. From Caminero et al., 1995.

GRÁFICO 2. INCIDENCIA DE TUBERCULOSIS POR 10⁵ h Y ÁREAS DE SALUD CON MAYOR AFECTACIÓN. CANARIAS, 2000-2016

Figure 2. Incidence of Tuberculosis in the Canary Islands between 2000-2016. Colors represent different islands; LZ: Lanzarote; FV: Fuerteventura; GC: Gran Canaria; TFE: Tenerife. Data obtained from Canarian Public Health Service <https://www3.gobiernodecanarias.org/sanidad/scs/content/4b07eed2-37d6-11e8-b144-0ff7a693f57f/Anuario2016.pdf>

TABLA 9.- TUBERCULOSIS. DISTRIBUCIÓN EN PORCENTAJE, SEGÚN FACTOR/SITUACIÓN DE RIESGO. CANARIAS 2010-2016

AÑO	VIH/SIDA	UDIs	ALCOHOLISMO
2000	12,3	11	8,4
2001	14,6	7,9	9,6
2002	9,9	8,7	12,3
2003	11,9	12,3	13,3
2004	6,5	8,3	11,5
2005	10,2	7,4	10,6
2006	7,7	6,0	9,5
2007	7,2	9,2	14,1
2008	5,5	4,2	2,1
2009	7,6	6,7	6,7
2010	6,7	6,7	7,7
2011	5,2	5,2	11,5
2012	10,9	11,5	12,8
2013	5,6	8	13
2014	8,8	11,3	18,8
2015	5,9	3,3	13,1
2016	8,8	4,8	8,8

Table 2. Percentage of TB cases for each risk factor for the Canary Islands between 2000-2016. UDI: parenteral drug users; ALCOHOLISMO: alcohol abuse. Data obtained from Canarian Public Health Service

<https://www3.gobiernodecanarias.org/sanidad/scs/content/4b07eed2-37d6-11e8-b144-0ff7a693f57f/Anuario2016.pdf>

Following the reviewer’s suggestion, we have included the following paragraphs in the manuscript:

Line 400: “On average, the outbreak has decreased since the last period showed a R_e close to 1. Other than its particular profile, we observed a median R_e of 6 [0.01 - 11 95% HPD], peaking at the beginning of the outbreak (Figure 4) and far from the value of 10-12 as the maximum number of secondary cases caused by an infected person per year reported for TB (35, 39). Of note, ongoing discussions exist regarding the estimates of infection period and R_e in TB; different factors (e.g., patient immunological status and social determinants) influence both parameters. Nevertheless, our mean values lie within the range of values observed for TB (40, 41), and there is no indication from phylodynamic analysis that the GC strain has a transmission advantage, at least in terms of a higher number of secondary cases generated of lengthened infection period.”

Line 537: “In summary, we describe four pieces of evidence supporting a lack of intrinsically higher transmissibility of the GC strain: i) Previous findings do not suggest any differential biological properties for the GC strain compared to control strains (11); ii) Epidemiological data support the existence of environmental factors associated with the transmission and, consequently, with the success of the GC strain and the development of the Gran Canaria

outbreak; iii) There exists little evidence of a similar expansion of the GC strain outside the Canary Islands, as revealed by our global genomic analysis and long-term molecular epidemiology studies in Spain; iv) Phylodynamic parameters associated with the GC strain lies within the expected range for TB. Overall, our data suggest that the success of the long-lived Gran Canaria outbreak-associated GC strain, and probably others, relates to ecological factors associated with founder effects linked to the host and social determinants of disease.”

Supplementary Notes:

“Epidemiological context of the Canary Islands

Gran Canaria, the most populated island in the archipelago, displayed a decreasing TB incidence from 32.2 cases per 100,000 inhabitants in 1988 to 29 in 1992, the year before the outbreak onset. The first years of the outbreak had an incidence of around 29 cases per 100,000 inhabitants (1–4). Meanwhile, the prevalence of Beijing (L2) strains increased from 5.5% of total cases in 1993 to 27.1% in 1999 (5), with a prevalence of 20.9% reported in 2014 (6). Alcoholism was the most important risk factor associated with TB before 1993, accounting for ~35% of total cases (2). During the first years of the outbreak (1993-1999), the only risk factor reported was HIV infection, accounting for 14% of cases (3). In 2000, the TB surveillance program was implemented in the Canary Islands by the Canarian Public Health Service, which prompted a reduction in TB incidence from 23.4 in 2000 to 6.3 in 2019 (<https://www3.gobiernodecanarias.org/sanidad/scs/listaImagenes.jsp?idDocument=fe9e3e94-fee-11e0-ab85-376c664a882a&idCarpeta=b25ca6dc-a9a4-11dd-b574-dd4e320f085c>).

3. The authors helpfully provide reasons and hypotheses for their findings for the GC strain outbreak, though the rationale is often missing. For example, In 485- why is it thought transmission was due to poor treatment adherence, rather than lack of diagnosis or other reasons? Were treatment-related mutations identified in the secondary strains? Specific statements should be supported with study or historical data especially to suggest effective TB control measures.

Regarding poor treatment adherence we referred to the index case, for which we linked epidemiological data. Treatment adherence and diagnostic delays are possible for the rest of cases, but we lack this data. No mutations associated with antimycobacterials were identified and therefore are not likely to exist (available catalogs of mutations for first-line drugs are comprehensive and should have detected any relevant mutation). We now make these two clarifications explicit in the main text. While it is not the primary purpose of this article, we believe our group shows the importance of early control of outbreaks and tackling the underlying causes (social, host, and pathogen determinants of disease) to avoid long-term consequences. In this case, the strain still causes 20% of cases in the Canary Islands thirty years after the initial case.

We have added the following paragraph in the discussion to clarify the point addressed by the reviewer:

Line 463: “By evaluating serial samples of the likely index case from different years, we identified another late and small secondary outbreak associated with this patient; the epidemiological information linked to this case suggests that poor treatment adherence could explain this patient’s most prolonged infectious period and, thus, the generation of that secondary outbreak.”

4. Additional limitations in terms of the patient-level data available for full evaluation of the transmissibility of the strains should be mentioned (SES, living situation, etc.).

Another additional limitation is the lack of robust epidemiological data for all cases included in the study; furthermore, we do not have information about living situations or socio-economic status.

We have added the following paragraph:

Line 512: "Additional limitations at the patient level include the lack of specific information regarding the socio-economic status of all cases and incomplete data regarding the living situation and other risk factors."

Minor comments

1. The submission should be revised for English grammar, as there are many syntactical errors, especially in the Introduction and discussion

We thank the reviewer for the suggestion; an English-speaking native has edited the manuscript. We hope it now matches the journal standard.

2. Please clarify In 148: "A representative number of samples from the Canary Islands"

We selected at least 15 samples per five years during the study period 1993-2014, to possess a balanced representation of the diversity of the outbreak. Nevertheless, not all samples had sufficient quality to obtain sequencing data, and not all the sequences belonged to the outbreak, considering they were assigned by molecular typing methods that only query a restricted proportion of the genome. As mentioned in question#1: The proportion of cases included in our analysis is comparable with other well-documented outbreaks (Folkvardsen et al., 2017), in which the number of cases allowed them to obtain valuable information about the origin and factors associated with the outbreak. In addition, we searched in the most extensive public repository for additional sequences of the outbreak harboring the defined SNP profiles to include more cases if available.

We have added the following paragraph in the materials section:

Line 139: "A representative number of samples were selected from the Canary Islands (i.e., at least 15 samples every 5 years during the study period 1993-2014), to ensure a balanced representation of the diversity of the outbreak. All samples were previously identified as part of the Gran Canaria outbreak by genotyping methods."

3. Where was WGS performed? In particular, please describe the infrastructure and sample referral/flow processes in the Canary Islands that contribute to the success of such an outbreak investigation.

The WGS was performed at the Instituto de Biomedicina de Valencia (Spain), and this information has been included in the Methods section.

We have included the following paragraph in the Supplementary Information:

“Study design

Canarian Hospitals performed TB diagnoses (GeneXpert and culture). The first cases were sent to the Instituto Aragonés de Ciencias de la Salud (Zaragoza, Spain), where they were typed by molecular methods (1). After the outbreak was declared, DNA extraction and RFLP typing were conducted in the Hospital Universitario de Gran Canaria Dr. Negrín between 1993-1996 (2). In the following years, inactivated samples from Canarian Hospitals were sent to the Instituto Aragonés de Ciencias de la Salud, for outbreak surveillance using spoligotyping. Epidemiological data were obtained retrospectively from the different Canarian Hospitals, Madrid, Zaragoza and Valencia.”

4. Consider providing the in-house R script in supplement

All scripts used in the methods section are available in our GitLab repository (<https://gitlab.com/tbgenomicsunit>). All the files needed for the script to run correctly have been placed in the repository. Nevertheless, we have included script names and links in the methods section of the manuscript.

5. Figure 5C: please adjust coloring so that locations and lines are clearer

Unfortunately we cannot adjust the colors of the Figure 5C, as it was constructed with Google Earth, which does not have the option of managing colors. Lines represent the same migration pattern represented in figure 5D in a 3D view. Since 2D and 3D views are complementary, we included both panels. To make the figure clearer for readers, we modified the legend of Figure 5, to indicate that both patterns are the same.

Figure 5 legend has been modified as follows:

“**Figure 5.** Dating, phylogeographic, and dispersal analysis. **A.** ML global phylogeny of L2 highlighting sublineages. Time of the most recent common ancestor (tMRCA) of L2.2. is indicated. Colored branches correspond to GC strain (red), African-related clade (orange), and African-Vietnamese clade (yellow). **B.** tMRCAs and origin of GC strain, closest nodes and oldest Chinese ancestor. Colors indicate the origin of either sample (labels) and the ancestor of different clades. **C-D.** The phylogeographic analysis results indicate routes of GC strain dispersion since the oldest ancestor. **C.** 3D view obtained with Google Earth (www.google.com/earth/). **D.** Map obtained using the *SPREAD3* software (https://rega.kuleuven.be/cev/ecv/software/Spread3_tutorial), lines are colored following the destination of the migration, and times from the oldest origin in China, to Vietnam and the Canary Islands are denoted.”

References

- Behr MA, Edelstein PH, Ramakrishnan L. Revisiting the timetable of tuberculosis. *BMJ*. 2018 Aug 23;362:k2738. doi: 10.1136/bmj.k2738. PMID: 30139910; PMCID: PMC6105930.
- Blower, S., Mclean, A., Porco, T. *et al.* The intrinsic transmission dynamics of tuberculosis epidemics. *Nat Med* 1, 815–821 (1995). <https://doi.org/10.1038/nm0895-815>
- Brooks-Pollock E, Danon L, Korthals Altes H, Davidson JA, Pollock AMT, et al. (2020) A model of tuberculosis clustering in low incidence countries reveals more transmission in the United Kingdom than the Netherlands between 2010 and 2015. *PLOS Computational Biology* 16(3): e1007687. <https://doi.org/10.1371/journal.pcbi.1007687>
- Caminero JA, Díaz F, Rodríguez de Castro F, Alonso JL, Daryanany RD, Carrillo T, Cabrera P. Epidemiología de la enfermedad tuberculosa en la isla de Gran Canaria [Epidemiology of tuberculosis in the Gran Canary Island]. *Med Clin (Barc)*. 1991 Jun

- 1;97(1):8-13. Spanish. PMID: 1857150.
- Camínero JA, Díaz F, Rodríguez de Castro F, Pavón JM, Esparza R, Cabrera P. The epidemiology of tuberculosis in Gran Canaria, Canary Islands, 1988–92: effectiveness of control measures. *Tuberc Lung Dis* 1995; 76:387–393.
- Camínero JA, Pena MJ, Campos MI, Samper S, Martín C. 1999. Epidemiología de la tuberculosis en la isla de Gran Canaria. Cuatro años de estudio poblacional mediante métodos de epidemiología convencional y por DNA fingerprinting. *Special monographic issue on Tuberculosis* (<http://www.sanipe.es/OJS/index.php/RESP/article/view/159/361>)
- Cancino-Muñoz I, López MG, Torres-Puente M, Villamayor LM, Borrás R, et al. 2022. Population-based sequencing of Mycobacterium tuberculosis reveals how current population dynamics are shaped by past epidemics *eLife* 11:e76605. <https://doi.org/10.7554/eLife.76605>
- Casali N, Broda A, Harris SR, Parkhill J, Brown T, Drobniowski F. 2016. Whole Genome Sequence Analysis of a Large Isoniazid-Resistant Tuberculosis Outbreak in London: A Retrospective Observational Study. *PLoS Med* 2016;13:e1002137.
- Casali, N., Nikolayevskyy, V., Balabanova, Y. et al. Evolution and transmission of drug-resistant tuberculosis in a Russian population. 2014. *Nat Genet* 46, 279–286. <https://doi.org/10.1038/ng.2878>
- Comín J, Madacki J, Rabanaque I, Zúñiga-Antón M, Ibarz D, et al. 2022. The MtZ Strain: Molecular Characteristics and Outbreak Investigation of the Most Successful Mycobacterium tuberculosis Strain in Aragon Using Whole-Genome Sequencing. *Frontiers in Cellular and Infection Microbiology* 12. 10.3389/fcimb.2022.887134
- Diel R, Schneider S, Meywald-Walter K, Ruf CM, Rüsç-Gerdes S, Niemann S. Epidemiology of tuberculosis in Hamburg, Germany: long-term population-based analysis applying classical and molecular epidemiological techniques. *J Clin Microbiol.* 2002 Feb;40(2):532-9. doi: 10.1128/JCM.40.2.532-539.2002. PMID: 11825968; PMCID: PMC153391.
- Emane AKA, Guo X, Takiff HE, Liu S. 2021. Highly transmitted M. tuberculosis strains are more likely to evolve MDR/XDR and cause outbreaks, but what makes them highly transmitted? *Tuberculosis*, Volume 129, <https://doi.org/10.1016/j.tube.2021.102092>.
- Frieden TR, Sherman LF, Maw KL, et al. A Multi-institutional Outbreak of Highly Drug-Resistant Tuberculosis: Epidemiology and Clinical Outcomes. *JAMA.* 1996; 276(15):1229–1235. doi:10.1001/jama.1996.03540150031027
- Kühnert D, Coscolla M, Brites D, Stucki D, Metcalfe J, et al. 2018. Tuberculosis outbreak investigation using phylodynamic analysis. *Epidemics* 25: 47-53. <https://doi.org/10.1016/j.epidem.2018.05.004>.
- Ma Y, Horsburgh CR, White LF, Jenkins HE. Quantifying TB transmission: a systematic review of reproduction number and serial interval estimates for tuberculosis. *Epidemiol Infect.* 2018 Sep;146(12):1478-1494. doi: 10.1017/S0950268818001760
- Pai, M., Behr, M., Dowdy, D. et al. Tuberculosis. *Nat Rev Dis Primers* 2, 16076 (2016). <https://doi.org/10.1038/nrdp.2016.76>
- Pena MJ, Camínero JA, Campos-Herrero MI, et al. 2003. Epidemiology of tuberculosis on Gran Canaria: a 4 year population study using traditional and molecular approaches. *Thorax* 2003;58:618-622.
- Pérez-Lago L, Herranz M, Comas I, Ruiz-Serrano MJ, Roa PL, Bouza E, et al. Ultrafast Assessment of the Presence of a High-Risk Mycobacterium tuberculosis Strain in a Population. *J Clin Microbiol* 2016;54:779–81.
- Pérez-Lago L, Campos-Herrero MI, Cañas F, Copado R, Sante L, Pino B, et al. A Mycobacterium tuberculosis Beijing strain persists at high rates and extends its geographic boundaries 20 years after importation. *Sci Rep* 2019;9:1–6.
- Pérez-Lago L, Navarro Y, Montilla P, Comas I, Herranz M, Rodríguez-Gallego C, et al. 2015. Persistent Infection by a Mycobacterium tuberculosis Strain That Was Theorized To Have Advantageous Properties, as It Was Responsible for a Massive Outbreak. *J Clin Microbiol* 53:3423–9.
- Ruddy MC, Davies AP, Yates MD, et al. 2004. Outbreak of isoniazid resistant tuberculosis in

- north London. *Thorax* 59:279-285. I
- Sanz B, Blasco T, for the ATBIM Project. 2007. The Union Variables associated with diagnostic delay in immigrant groups with tuberculosis in Madrid . *Int J Tuberc Lung DIS* 11(6):639–646
- Seminario A, Anibarro,L, Sabriá J, García-Clemente MM, Sánchez-Montalván A, et al. 2021. Estudio del retraso diagnóstico de la tuberculosis en España. *Archivos de Bronconeumología* 57: 440-442. DOI: [10.1016/j.arbres.2020.09.003](https://doi.org/10.1016/j.arbres.2020.09.003)
- Valway SE, Sanchez MP, Shinnick TF, Orme I, Agerton T, et al. 1998. An outbreak involving extensive transmission of a virulent strain of *Mycobacterium tuberculosis*. *N Engl J Med.*;338(10):633-9. doi: 10.1056/NEJM199803053381001. Erratum in: *N Engl J Med* 1998 Jun 11;338(24):1783. PMID: 9486991.
- Walker TM, Lalor MK, Broda A, Ortega LS, Morgan M, et al. 2014. Assessment of *Mycobacterium tuberculosis* transmission in Oxfordshire, UK, 2007-12, with whole pathogen genome sequences: an observational study. *The Lancet. Respiratory Medicine* 2:285–292. DOI: [https://doi.org/10.1016/S2213-2600\(14\)70027-X](https://doi.org/10.1016/S2213-2600(14)70027-X), PMID: 24717625

November 10, 2022

Dr. Mariana G Lopez
Instituto de Biomedicina de Valencia
Valencia
Spain

Re: Spectrum02826-22R1 (**Deciphering the tangible spatio-temporal spread of a 25-years tuberculosis outbreak boosted by social determinants**)

Dear Dr. Mariana G Lopez:

Your manuscript revision has been reviewed by two experts in the field. Although most of the original issues have been addressed, modifications are still needed. Reviewer 1 makes an excellent point about differentiating between the description of an outbreak (cluster of cases) versus a lineage (bacterial strain/genotype). Please make these changes for clarity throughout the paper.

Link Not Available

Thank you for submitting your manuscript to Microbiology Spectrum.

Sincerely,

Shannon Manning

Journals Department
Reviewer comments:

Reviewer #1 (Comments for the Author):

The authors edits to the second point are appreciated.
Regarding the word outbreak however, the authors conceded in their rebuttal that the word is used to describe different things. But instead of removing the word outbreak when talking about a clade of bacteria, they have added this word and introduced a second definition. This is now harder to read.

In the second sentence of the introduction, the authors write: "Outbreaks, defined as the concentration of an abnormal number of disease cases in space and time, are widespread in TB". In their rebuttal they mention adding a sentence: "We added a sentence to clarify the definition of the outbreak by using a phylogenetic approach". So there is an epidemiologic definition of an outbreak at the beginning and a phylogenetic definition later on.

The authors need to choose one definition and use it consistently. Otherwise, the paper tells us that there is an outbreak (epidemiological event) on GC and the outbreak (bacterium) went to Switzerland, as confirmed by a single secondary case. Can they not use outbreak in the conventional manner, to describe an epidemiological concentration of cases, and use clade or lineage or something different to describe their bacterial data?

Reviewer #3 (Comments for the Author):

The authors did an excellent job in considering and addressing all comments. The revisions have greatly improved the submission and I have no further comments.

Staff Comments:

Preparing Revision Guidelines

Please return the manuscript within 60 days; if you cannot complete the modification within this time period, please contact me. If you do not wish to modify the manuscript and prefer to submit it to another journal, please notify me of your decision immediately so that the manuscript may be formally withdrawn from consideration by Microbiology Spectrum.

PLEASE NOTE - Line numbers in the below refer to modified version of the revised edition.

Reviewer #1 (Comments for the Author):

The author's edits to the second point are appreciated.

Regarding the word outbreak however, the authors conceded in their rebuttal that the word is used to describe different things. But instead of removing the word outbreak when talking about a clade of bacteria, they have added this word and introduced a second definition. This is now harder to read.

In the second sentence of the introduction, the authors write: "Outbreaks, defined as the concentration of an abnormal number of disease cases in space and time, are widespread in TB". In their rebuttal they mention adding a sentence: "We added a sentence to clarify the definition of the outbreak by using a phylogenetic approach". So there is an epidemiologic definition of an outbreak at the beginning and a phylogenetic definition later on.

The authors need to choose one definition and use it consistently. Otherwise, the paper tells us that there is an outbreak (epidemiological event) on GC and the outbreak (bacterium) went to Switzerland, as confirmed by a single secondary case. Can they not use outbreak in the conventional manner, to describe an epidemiological concentration of cases, and use clade or lineage or something different to describe their bacterial data?

We apologise for adding extra difficulties in reading the manuscript. Unfortunately, lineage and clade are both terms already defined and widely used in tuberculosis. There are eight *M. tuberculosis* well-described lineages, and the GC strain particularly belongs to lineage 2. In addition, clade is a term used and well-defined in phylogenies. In our opinion, both terms are not entirely correct to describe the group of related genotypes we are trying to define.

In this version we now expand the definition included in the introduction to clarify some concepts for a broader audience. Tuberculosis outbreaks -or the increase of TB cases in a particular region during a particular period of time- are caused or ascribed to a single genotype or strain (as is the case of C2 outbreak in Denmark, the M strain in Argentina) (1,2). The common use of the term "outbreak" in specialised TB literature, and also by CDC (<https://www.cdc.gov/tb/education/ssmodules/pdfs/modules9-508.pdf>), refers to increase number of TB cases infected by a particular genotype, as is the case presented in the manuscript. We used the term "outbreak" in this sense in the whole manuscript.

Following the reviewer suggestions, and with the aim of making the read more clear and precise, we made the following changes in the manuscript:

- Outbreak definition was expanded in the introduction:

Line 115: "Outbreaks, defined as the concentration of an abnormal number of disease cases in space and time, are widespread in TB **and ascribed to a single genotype or strain (1).**"

- The phylogenetic definition included in the Results section was deleted. We updated the next paragraph to clearly explain the use of "GC outbreak" throughout the manuscript. The revised literature about TB outbreaks uses the term "outbreak" referring to cases infected by the same genotype, or to isolates of the same genotype (2–4)

Line 312: "This clade likely represents the most precise genomic delimitation of the outbreak based on WGS data and is referred to as "GC outbreak" hereafter, in order to indicate the outbreak caused by this particular genotype (GC strain) ."

- When referring to expansion or spread we use the term "GC strain" avoiding the use of outbreak expansion or outbreak spread outside the Canarian Archipelago
- When referring to evolution and ancestry, we also use the term "GC strain". In this sense we "Tracked the GC strain over centuries" and identified the "GC strain's oldest ancestor in China" or defined the time of the most common ancestor (tMRCA) of the closest GC strain's clades.
- In addition we changed Figures 1 and 2 in accordance to this definition.

We hope these changes help streamline the read and better match the reviewer requirements. We understand the differences between the epidemiological definition of a disease outbreak, which only refer to the abnormal increase of disease cases (as in measles); but in TB all outbreaks are associated with a specific strain, in our case studied by WGS -which is more precise than other molecular typing methods-. Our approach makes it more difficult to differentiate epidemiological outbreak from outbreak genomic data, since we analysed mainly the genomic data of the strain causing the outbreak.

1. Folkvardsen DB, Norman A, Andersen ÅB, Michael Rasmussen E, Jelsbak L, Lillebaek T. 2017. Genomic Epidemiology of a Major Mycobacterium tuberculosis Outbreak: Retrospective Cohort Study in a Low-Incidence Setting Using Sparse Time-Series Sampling. *J Infect Dis* 216:366–374.
2. Li H, Liu C, Liang M, Liu D, Zhao B, Shi J, Zhao Y, Ou X, Zhang G. 2021. Tuberculosis Outbreak in an Educational Institution in Henan Province, China. *Front Public Health* 9.
3. Smit PW, Vasankari T, Aaltonen H, Haanperä M, Casali N, Marttila H, Marttila J, Ojanen P, Ruohola A, Ruutu P, Drobniewski F, Lyytikäinen O, Soini H. 2015. Enhanced tuberculosis outbreak investigation using whole genome sequencing and IGRA. *Eur Respir J* 45:276–279.
4. Walker TM, Monk P, Smith EG, Peto TE. 2013. Contact investigations for outbreaks of Mycobacterium tuberculosis: advances through whole genome sequencing. *Clin Microbiol Infect* 19:796–802.

Reviewer #3 (Comments for the Author):

The authors did an excellent job in considering and addressing all comments. The revisions have greatly improved the submission and I have no further comments.

We thank the reviewer for their positive assessment of our revision.

January 16, 2023

Dr. Mariana G Lopez
Instituto de Biomedicina de Valencia
Valencia
Spain

Re: Spectrum02826-22R2 (**Deciphering the tangible spatio-temporal spread of a 25-years tuberculosis outbreak boosted by social determinants**)

Dear Dr. Mariana G Lopez:

Thank you for submitting your manuscript to Microbiology Spectrum. Before your paper can be accepted, please address the inconsistencies and edits requested by Reviewer 2. In your revision, please provide (1) point-by-point responses to the issues raised by the reviewers as file type "Response to Reviewers," not in your cover letter, and (2) a PDF file that indicates the changes from the original submission (by highlighting or underlining the changes) as file type "Marked Up Manuscript - For Review Only". Please use this link to submit your revised manuscript - we strongly recommend that you submit your paper within the next 60 days or reach out to me. Detailed instructions on submitting your revised paper are below.

Link Not Available

Sincerely,

Shannon Manning

Journals Department
Reviewer comments:

Reviewer #1 (Comments for the Author):

Revisions appreciated.

Reviewer #3 (Comments for the Author):

The authors' points are well taken regarding the terminology used to define outbreaks in the TB field and the authors helpfully define these terms in the context of TB. The text is improved, especially through the removal of the text regarding the "phylogenetic definition," and there are only a few minor inconsistencies and errors to still correct:

- In the Results section the authors state that "[the clade is] referred to as "GC outbreak" hereafter", yet the "GC outbreak" is references multiple times prior in the text. Please correct this inconsistency.
- In edited text, change "track" to "tracked" to preserve common tense

- S1 Table- Please revise or delete column "J"
- Please give units for sequencing depth
- S2 Table- Correct "SPN" to "SNP"
- Define all acronyms in all supplemental tables

Staff Comments:

Preparing Revision Guidelines

Please return the manuscript within 60 days; if you cannot complete the modification within this time period, please contact me. If you do not wish to modify the manuscript and prefer to submit it to another journal, please notify me of your decision immediately so that the manuscript may be formally withdrawn from consideration by Microbiology Spectrum.

Reviewer comments:

We want to thank both reviewers and editors for all their comments and suggestions which significantly improved our work. We addressed all issues raised and have marked the changes in the file named as "Lopez_etal_GCoutbreak_R3_marked.pdf"

Reviewer #1 (Comments for the Author):

Revisions appreciated.

Reviewer #3 (Comments for the Author):

The authors' points are well taken regarding the terminology used to define outbreaks in the TB field and the authors helpfully define these terms in the context of TB. The text is improved, especially through the removal of the text regarding the "phylogenetic definition," and there are only a few minor inconsistencies and errors to still correct:

-In the Results section the authors state that "[the clade is] referred to as "GC outbreak" hereafter", yet the "GC outbreak" is references multiple times prior in the text. Please correct this inconsistency.

Response: Thanks to the reviewer for the deep revision of our manuscript, we modified all the "GC outbreak" terms before the Results section to be consistent with the definition.

-In edited text, change "track" to "tracked" to preserve common tense

Response: We changed "track" by "tracked" in the introduction.

-S1 Table- Please revise or delete column "J"

-Please give units for sequencing depth

Response: Column "J" was deleted, thanks for noticing the mistake.

Median sequencing depth is commonly indicated as a number plus an "X", without units; this is useful data in all sequencing studies to know how confident is the variant calling. This value indicates the median number of times each position was read or sequenced, we added "reads" as units and a brief explanation of "median depth" at the bottom of the table.

-S2 Table- Correct "SPN" to "SNP"

Response: Done

-Define all acronyms in all supplemental tables

Response: Done, all acronyms were defined at the bottom of each table.

January 18, 2023

Dr. Mariana G Lopez
Instituto de Biomedicina de Valencia
Valencia
Spain

Re: Spectrum02826-22R3 (**Deciphering the tangible spatio-temporal spread of a 25-years tuberculosis outbreak boosted by social determinants**)

Dear Dr. Mariana G Lopez:

Thank you for addressing the comments by the two reviewers. Your manuscript has been accepted, and I am forwarding it to the ASM Journals Department for publication. You will be notified when your proofs are ready to be viewed.

Congratulations and thank you for submitting your paper to Spectrum.

Sincerely,

Shannon Manning
Editor, Microbiology Spectrum
